# Symmetric Divergence and Normalized Similarity: A Unified Topological Framework for Representation Analysis

**Yan Wang**                                                                    *wangyan1@cuhk.edu.cn*
*School of Data Science*
*The Chinese University of Hong Kong, Shenzhen*

**Tianyang Hu**                                                                  *hutianyang@cuhk.edu.cn*
*School of Data Science*
*The Chinese University of Hong Kong, Shenzhen*

**Reviewed on OpenReview:** *https://openreview.net/forum?id=pGgJ9qB2Io*

## Abstract

Topological Data Analysis (TDA) offers a principled, intrinsic lens for comparing neural representations. However, existing paired topological divergences (e.g., RTD) are limited by heuristic asymmetry and, more critically, unbounded scores that depend on sample size, hindering reliable cross-scenario benchmarking. To address these challenges, we develop a unified topological toolkit serving two complementary needs: fine-grained structural diagnosis and robust, standardized evaluation. First, we complete the RTD framework by introducing **Symmetric Representation Topology Divergence (SRTD)** and its efficient variant **SRTD-lite**. Beyond resolving the theoretical asymmetry of prior variants, SRTD consolidates diagnostic information into a single, comprehensive cross-barcode signature. This allows for precise localization of structural discrepancies and serves as an effective optimization objective without the overhead of dual directional computations. Second, to enable reliable benchmarking across heterogeneous settings, we propose **Normalized Topological Similarity (NTS)**. By measuring the rank correlation of hierarchical merge orders, NTS yields a scale-invariant metric bounded between -1 and 1, effectively overcoming the scale and sample-dependence of unnormalized divergences. Experiments across synthetic and real-world deep learning settings demonstrate that our toolkit captures functional shifts in CNNs missed by geometric measures and robustly maps LLM genealogy even under distance saturation, offering a rigorous, topology-aware perspective that complements measures like CKA.

## 1 Introduction

Understanding the internal representations of neural networks is a central challenge in deep learning, crucial for interpreting their behavior and improving their design. In representation analysis, it is common to compare activations obtained from the *same* collection of inputs—for instance, from different models or different layers—which induces a natural sample-wise correspondence between representations (Klabunde et al., 2025; Lenc & Vedaldi, 2015; Li et al., 2016; Chen et al., 2023; Bansal et al., 2021; Csiszárik et al., 2021). Early approaches focused on subspace-based comparisons, most notably Canonical Correlation Analysis (CCA) and its variants such as SVCCA (Raghu et al., 2017) and PWCCA (Morcos et al., 2018). These methods, however, can be overly permissive, as they remain invariant under arbitrary invertible linear transformations. To address this, Centered Kernel Alignment (CKA) (Kornblith et al., 2019) was proposed and has since become a widely adopted tool. By comparing centered Gram matrices, CKA yields a normalized similarity score that facilitates comparison across diverse settings and is robust to fundamental geometric transformations.

---

Code is available at: `https://github.com/frankwy505/SRTD-NTS`.

While geometric analysis dominates the field, Topological Data Analysis (TDA) offers a complementary perspective by probing the intrinsic shape of data. Using tools like persistent homology (Barannikov, 1994; Carlsson et al., 2004), this line of approaches examines how the fundamental topological structure of the data—from simple clusters to complex loops and voids—is formed and evolves across a continuous range of scales.

Although certain topological methods effectively evaluate individual network units at a microscopic level (Zhao & Zhang, 2022), existing macroscopic approaches for comparing representations face distinct limitations regarding their applicability. Methods such as Geometry Score (Khrulkov & Oseledets, 2018) and IMD (Tsitsulin et al., 2019) are highly general and do not require a one-to-one correspondence between representations. While flexible, they fail to leverage the valuable pairing information inherent in comparing neural network layers, often resulting in lower discriminative power. Conversely, approaches that do analyze distributional topology often strictly require the point clouds to reside in the same ambient space (Kynkäänniemi et al., 2019; Barannikov et al., 2021b), severely limiting their scope.

A significant breakthrough in bridging this gap is Representation Topology Divergence (RTD) (Barannikov et al., 2021a) and its scalable variant, RTD-lite (Tulchinskii et al., 2025). These methods successfully utilize the one-to-one correspondence between data points without requiring them to share the same ambient space, making them powerful tools for representation analysis and optimization (Trofimov et al., 2023).

Despite this progress, the RTD framework still faces two key limitations that hinder its broader adoption. First, its theoretical underpinnings remain incomplete: the commonly used "symmetric" RTD is defined as a brute-force average of two directional values, $RTD(w, \tilde{w})$ and $RTD(\tilde{w}, w)$, which can differ dramatically (Figure 4f) without a clear explanation of when and why such asymmetry should occur. Moreover, its dual variant, Max-RTD, proposed by Trofimov et al. (2023) to enrich gradient information, has not been fully characterized theoretically, leaving its role and relationship to RTD ambiguous. Second, and more fundamentally, divergence-style topological measures are *not normalized*. The outputs of RTD and RTD-lite are unbounded positive numbers that depend strongly on the number of sample points and the intrinsic distance scale, making cross-scenario comparison difficult and interpretability elusive. These two limitations correspond exactly to the dual needs our toolkit targets: fine-grained diagnosis and robust cross-scenario evaluation.

Our overarching goal is to develop a cohesive representation analysis toolkit from a topological perspective. To this end, we introduce two complementary components that address different but equally essential needs: fine-grained structural diagnosis and standardized, cross-scenario similarity evaluation.

- For structural diagnosis: SRTD and SRTD-lite. We complete the RTD framework by introducing Symmetric Representation Topology Divergence (SRTD) and its lightweight variant. While preserving the fine-grained diagnostic power and optimization utility of the original RTD family, SRTD offers a crucial advancement in efficiency and interpretability. Unlike prior directional variants that require dual computations, SRTD captures the total topological discrepancy in a single, symmetric cross-barcode. This not only provides a unified theoretical link between RTD and Max-RTD but also serves as a more efficient objective for optimization with comparable quality.

- For standardized evaluation: NTS. To overcome the limitations of divergence-based measures—specifically their unbounded nature and dependence on sample size—we introduce Normalized Topological Similarity (NTS). By leveraging intrinsic rank-based normalization to compare hierarchical merge orders, NTS yields a standardized score bounded between -1 and 1. This formulation ensures consistency across varying dataset sizes and scenarios, enabling robust benchmarking where unnormalized divergences would be incomparable, and offering a topology-aware complement to geometric baselines like CKA.

The rest of this paper is structured as follows. In Section 2, we introduce notation and review persistent homology as well as the RTD/RTD-lite framework. In Section 3, we present SRTD and SRTD-lite and establish their theoretical relationships with RTD and Max-RTD. In Section 4, we propose Normalized Topological Similarity (NTS) and discuss its algorithmic definition and basic properties. In Section 5,

we evaluate our toolkit on synthetic hierarchical shifts, CNN layer-wise analysis, and LLM case studies. Computational complexity and scalability are analyzed in Section 6. Finally, Section 7 concludes and outlines limitations and future directions.

Additional materials are deferred to the appendix, including (i) full algorithmic details and proofs for SRTD/SRTD-lite and NTS (Appendix A–B), (ii) supplementary barcode visualizations and qualitative interpretations, including ultra-long barcode cases and query-level diagnostics (Appendix H, J), (iii) an RSA baseline analysis using the full distance matrices (Appendix G), and (iv) extended experimental setups and additional analyses/heatmaps (Appendix E, F, C, I.2).

## 2 Preliminaries: Persistent Homology and Representation Topology Divergence

In the context of representation analysis, we consider a dataset of $n$ samples processed by a neural network. Let $X = \{x_1, \ldots, x_n\} \subset \mathbb{R}^d$ and $X' = \{x'_1, \ldots, x'_n\} \subset \mathbb{R}^{d'}$ be two sets of representations (e.g., activations from different layers or models) associated with these samples. We treat these representations as finite sets or **point clouds**, denoted as $P$ and $P'$, equipped with a pairwise **dissimilarity measure** (e.g., Euclidean distance or Cosine dissimilarity).

Accordingly, we represent the data through their pairwise dissimilarity matrices $w, \tilde{w} \in \mathbb{R}^{n \times n}$(We assume $w$ and $\tilde{w}$ are symmetric and nonnegative with zero diagonal: $w_{ij} = w_{ji} \geq 0$ and $w_{ii} = 0$ (and similarly for $\tilde{w}$).), where $w_{ij} = \text{dissim}(x_i, x_j)$. Since both matrices arise from the same set of samples, this induces a natural one-to-one correspondence between the indices of $w$ and $\tilde{w}$. We define $\min(w, \tilde{w})$ and $\max(w, \tilde{w})$ as the element-wise minimum and maximum of the two matrices, respectively.

To understand the topological structure of these point clouds, we employ persistent homology. The process can be intuitively understood as follows: for a given point cloud $P$ with distance matrix $w$, we construct a sequence of simplicial complexes, known as the Vietoris-Rips filtration (Hausmann, 1995), indexed by a proximity parameter $\alpha$. As $\alpha$ increases from zero, edges are added between points with distance less than or equal to $\alpha$. When a set of $n$ points are all mutually connected, the $(n-1)$-simplex they span is filled in (e.g., three points form a filled triangle). This growing complex is denoted as $R_\alpha(\mathcal{G}^w)$.Formally, the Vietoris-Rips complex $R_\alpha(\mathcal{G}^w)$ consists of all finite subsets (simplices) of $X$ with diameter at most $\alpha$: a simplex $\sigma = \{x_{i_0}, \ldots, x_{i_k}\}$ is included in $R_\alpha$ if and only if $w_{i_j i_l} \leq \alpha$ for all pairs in $\sigma$.

During this filtration process, topological features—such as connected components ($H_0$), cycles ($H_1$), and voids ($H_2$)—appear and disappear. We track the lifespan of each feature by recording its birth and death values as an interval $[b, d]$ (Barannikov, 1994). The collection of these intervals is known as **barcodes** (Carlsson et al., 2004), which serves as a topological signature of the point cloud. The computation of persistent homology operates directly on the distance matrix.

**RTD** A set of barcodes characterizes one point cloud. To compare two, Representation Topology Divergence (RTD) (Barannikov et al., 2021a) introduced an auxiliary matrix $M_{min}$ (Matrix 1b) constructed from $w$, $\tilde{w}$, and $\min(w, \tilde{w})$. The resulting barcode captures the differences in the evolution of topological features between an individual point cloud and the composite structure formed by their union, which is derived from the $\min(w, \tilde{w})$ matrix. The length of a barcode interval in this context quantifies the discrepancy between when a feature forms in $w$ (or $\tilde{w}$) versus when it forms in $\min(w, \tilde{w})$.

We define $RTD(w, \tilde{w})$ as the sum of the lengths of all barcodes computed from $M_{min}$ (Matrix 1b). By swapping the roles of $w$ and $\tilde{w}$, we can similarly compute $RTD(\tilde{w}, w)$. To ensure symmetry, the final divergence is typically defined as their average:$RTD(P, P') = \frac{RTD(w, \tilde{w}) + RTD(\tilde{w}, w)}{2}$ Subsequently, Trofimov et al. (2023) noted that a dual variant, which we term Max-RTD, can be defined by using an auxiliary matrix $M_{max}$ (Matrix 1c) based on $w$, $\tilde{w}$, and $\max(w, \tilde{w})$. However, the properties of this variant were not deeply investigated in their work. The symmetric versions of Max-RTD are defined analogously by averaging the two directional computations.

**RTD-lite** To alleviate the computational burden of high-dimensional persistent homology, RTD-lite (Tulchinskii et al., 2025) focuses exclusively on 0-dimensional features—specifically, the merging of con-

nected components. Its core efficiency derives from the equivalence between the topological divergence and the difference of Minimum Spanning Tree (MST) weights. Specifically, the directional divergence is defined as $RTD\_lite(w, \tilde{w}) = MST(w) - MST(\min(w, \tilde{w}))$. This MST-based formulation provides a scalable and computationally feasible framework for large-scale representation analysis.

**Notation for Vietoris-Rips Complexes** To streamline the following sections, we establish notation for the key Vietoris-Rips complexes used in our analysis. Recall that these are constructed based on a proximity parameter, $\alpha$, which acts as a distance threshold for connecting points. For any given threshold $\alpha$, we denote the complexes generated from the distance matrices $w$ and $\tilde{w}$ as $R_\alpha(\mathcal{G}^w)$ and $R_\alpha(\mathcal{G}^{\tilde{w}})$, respectively. The complexes derived from the element-wise minimum and maximum matrices have a crucial relationship to these: at the same scale $\alpha$, $R_\alpha(\mathcal{G}^{\min(w,\tilde{w})})$ is the union of the individual complexes $(R_\alpha(\mathcal{G}^w) \cup R_\alpha(\mathcal{G}^{\tilde{w}}))$, while $R_\alpha(\mathcal{G}^{\max(w,\tilde{w})})$ is their intersection $(R_\alpha(\mathcal{G}^w) \cap R_\alpha(\mathcal{G}^{\tilde{w}}))$.

$$
\begin{pmatrix} \max(w,\tilde{w}) & (\max(w,\tilde{w})^+)^T & 0 \\ \max(w,\tilde{w})^+ & \min(w,\tilde{w}) & \infty \\ 0 & \infty & 0 \end{pmatrix}
\qquad
\begin{pmatrix} w & (w^+)^T & 0 \\ w^+ & \min(w,\tilde{w}) & \infty \\ 0 & \infty & 0 \end{pmatrix}
\qquad
\begin{pmatrix} \max(w,\tilde{w}) & (\max(w,\tilde{w})^+)^T & 0 \\ \max(w,\tilde{w})^+ & w & \infty \\ 0 & \infty & 0 \end{pmatrix}
$$

$$\text{(a) } M_\text{sym} \qquad\qquad\qquad \text{(b) } M_\text{min} \qquad\qquad\qquad \text{(c) } M_\text{max}$$

Figure 1: The three key auxiliary matrices. For any matrix $M$, $M^+$ is obtained by replacing its upper triangular part with infinity.

## 3 Symmetric Representation Topology Divergence (SRTD)

In practice, we observe a complementary phenomenon between RTD and Max-RTD (shown in Figure 4f). When $RTD(w, \tilde{w}) > RTD(\tilde{w}, w)$, we consistently find that $Max\text{-}RTD(w, \tilde{w}) < Max\text{-}RTD(\tilde{w}, w)$. This suggests that the topological structural differences between $R_\alpha(\mathcal{G}^w) \cup R_\alpha(\mathcal{G}^{\tilde{w}})$ and $R_\alpha(\mathcal{G}^w) \cap R_\alpha(\mathcal{G}^{\tilde{w}})$ seem to be the core reason for the asymmetry in RTD. Therefore, we propose to directly measure this difference as the Symmetric Representation Topology Divergence (SRTD) of $P$ and $P'$.

**Definition 3.1** (SRTD). For two point clouds $P$ and $P'$ with a one-to-one correspondence, the distance matrix of their auxiliary graph $\hat{\mathcal{G}}'_{sym}$ is $M_{sym}$ (Matrix 1a). The sum of the lengths of its persistent homology barcodes is defined as $SRTD(P, P')$ (see Algorithm 3). Its chain complex is homotopy equivalent to the mapping cone of the inclusion map $f' : C_*(R_\alpha(\mathcal{G}^w) \cap R_\alpha(\mathcal{G}^{\tilde{w}})) \to C_*(R_\alpha(\mathcal{G}^w) \cup R_\alpha(\mathcal{G}^{\tilde{w}}))$.

To significantly reduce computational complexity, Tulchinskii et al. (2025) proposed **RTD-lite** by simplifying topological analysis to MST-based calculations (as detailed in Section 2). Following this lightweight framework, we extend the principle to the intersection structure $\max(w, \tilde{w})$ to formally define **Max-RTD-lite**. This completes the family alongside our proposed **SRTD-lite**, which directly quantifies the divergence between the composite union and intersection structures by comparing their respective MST weights.

**Definition 3.2** (SRTD-lite). By comparing the minimum spanning trees of $\min(w, \tilde{w})$ and $\max(w, \tilde{w})$ through Algorithm 4, we can obtain a series of barcodes. We define the sum of the lengths of these barcodes as $SRTD\text{-}lite(w, \tilde{w})$.

### 3.1 Mathematical Properties

SRTD, RTD, and Max-RTD satisfy some elegant mathematical properties. The mapping cones corresponding to their auxiliary graphs (defined by the dissimilarity matrices $M_{sym}$ 1a, $M_{min}$ 1b, and $M_{max}$ 1c) fit into the following long exact sequence:

$$\cdots \to H_n(R_\alpha(\mathcal{G}^w), R_\alpha(\mathcal{G}^{\max(w,\tilde{w})})) \xrightarrow{\gamma_n} H_n(R_\alpha(\mathcal{G}^{\min(w,\tilde{w})}), R_\alpha(\mathcal{G}^{\max(w,\tilde{w})}))$$

$$\xrightarrow{\beta_n} H_n(R_\alpha(\mathcal{G}^{\min(w,\tilde{w})}), R_\alpha(\mathcal{G}^w)) \xrightarrow{\delta_n} H_{n-1}(R_\alpha(\mathcal{G}^w), R_\alpha(\mathcal{G}^{\max(w,\tilde{w})})) \xrightarrow{\gamma_{n-1}} \cdots$$

**Theorem 3.3.** *For any dimension $i$, point clouds $P, P'$ and distance matrices $w, \tilde{w}$, the three divergences satisfy the following relationship:*

$$Max\text{-}RTD_i(w, \tilde{w}) + RTD_i(w, \tilde{w}) - SRTD_i(w, \tilde{w}) = \int_0^\infty (\dim(\ker(\gamma_i)) + \dim(\ker(\gamma_{i-1})))d\alpha$$

By swapping the positions of $w$ and $\tilde{w}$ in Theorem 3.3, we obtain a similar equality. We denote $RTD_i(w, \tilde{w}) + Max\text{-}RTD_i(w, \tilde{w})$ as $minmax(w, \tilde{w})$, and $RTD_i(\tilde{w}, w) + Max\text{-}RTD_i(\tilde{w}, w)$ as $minmax(\tilde{w}, w)$. Both are strictly greater than SRTD, but in our experiments, we find this gap to be very small, as shown in Figure 4e.

The introduction of SRTD provides a more mathematically elegant framework for understanding the RTD family. Within this framework, the asymmetric measures $minmax(w, \tilde{w})$ and $minmax(\tilde{w}, w)$ can be decomposed into a large, shared symmetric component, $SRTD(w, \tilde{w})$, and smaller, 'private' components. These private components correspond to topological features unique to the individual filtrations of $\mathcal{G}^w$ or $\mathcal{G}^{\tilde{w}}$ relative to the bounding filtrations of $\mathcal{G}^{\min(w, \tilde{w})}$ and $\mathcal{G}^{\max(w, \tilde{w})}$. This decomposition reveals that the asymmetry in the original RTD arises from these small, private feature sets, making the source of the divergence interpretable. The relationship becomes even more direct and elegant in the lite version:

**Corollary 3.4.** $Max\text{-}RTD\text{-}lite(w, \tilde{w}) + RTD\text{-}lite(w, \tilde{w}) = SRTD\text{-}lite(w, \tilde{w})$

**Corollary 3.5.** $Max\text{-}RTD\text{-}lite(P, P') \geq SRTD\text{-}lite(P, P') \geq RTD\text{-}lite(P, P')$

Together, Theorem 3.3 and Corollary 3.4, 3.5 provide a clear theoretical basis for a consistent pattern observed in our experiments: when plotting the divergence curves for either the full or lite families, the Max-RTD curve is always highest, the RTD curve is lowest, and the SRTD curve lies in between (as shown in Figure 4b). For the lite versions, Corollary 3.5 proves this hierarchical ordering is strict, which explains why the SRTD-lite curve appears perfectly centered between the other two. While the relationship for the full RTD family is more complex, this structure holds empirically, positioning SRTD as a balanced, median measure of topological divergence.

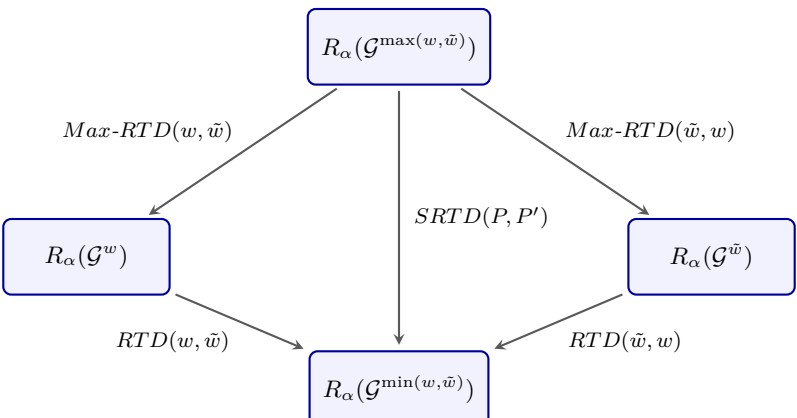

Figure 2: Conceptual relationship between SRTD, RTD, and Max-RTD.

# 4 Normalized Topological Similarity (NTS)

## 4.1 Motivation

As the second component of our unified topological toolkit, we now turn from diagnostic divergences to a normalized similarity for robust cross-scenario comparison.

Building on the cross-barcode construction introduced in RTD and its variants, we obtain a principled way to compare the topology of paired representations without requiring a shared ambient space (Barannikov et al., 2021a; Tulchinskii et al., 2025; Hu et al., 2023). Importantly, this construction yields fine-grained

and interpretable barcodes (not just a single scalar), making RTD/SRTD particularly useful for structural diagnosis; when optimization is desired, the associated divergence can also serve as a natural loss term.

However, for *general similarity analysis*—especially when comparisons must be made across layers, models, datasets, or experimental pipelines—the reliance on summing barcode lengths introduces fundamental limitations. First, the resulting values are *unnormalized*: they can grow with the number of samples and depend strongly on the distance scale, making scores hard to interpret and difficult to compare across scenarios. Existing practice partially mitigates this by rescaling distances (e.g., dividing by a high quantile such as the 0.9-quantile) before computing RTD/RTD-lite (Barannikov et al., 2021a; Tulchinskii et al., 2025), but this is inherently heuristic and cannot fully eliminate scale effects across heterogeneous settings.

Second, the sum of barcode lengths sometimes can be dominated by a few "ultra-long" intervals (Figure 20a). In practice, these intervals sometimes come from a small number of corresponding sample pairs, whose contributions account for a large fraction of the total divergence. This can be undesirable: a divergence meant to summarize *global* dissimilarity becomes overly sensitive to a handful of outlier pairs, potentially obscuring the overall structural relationship. This motivates a complementary goal: a *normalized*, scale-robust similarity that captures hierarchical structure comparably across settings.

Several TDA pipelines turn persistence diagrams into objects that support inner-product based comparisons, e.g., via positive-definite kernels or stable vectorizations such as persistence landscapes and persistence images (Reininghaus et al., 2015; Kusano et al., 2016; Bubenik, 2015; Adams et al., 2017). These approaches treat each diagram as an unordered summary and are therefore agnostic to sample-wise correspondence. To our knowledge, a normalized topological similarity that explicitly exploits this pairing is still missing.

## 4.2 Method: Capturing Merge-Order Similarity

**From RSA to NTS.** A core idea in representational similarity analysis (RSA) is to compare representations through their *representational dissimilarity matrices* (RDMs), i.e., the full set of pairwise dissimilarities between the same inputs (the most basic choice is simply the pairwise distance matrix), and to compare two RDMs by how similarly they order these pairwise relations (Kriegeskorte et al., 2008; Nili et al., 2014). Rank-based comparisons are attractive because they yield a normalized score and reduce sensitivity to global rescalings or other monotone distortions of dissimilarities. We adopt this RSA principle, but replace the *full* RDM vectorization with a topology-aware summary that focuses on 0D connectivity events (merge order) rather than all pairwise magnitudes. Accordingly, we use Spearman's rank correlation, defined as the Pearson correlation of rank-transformed vectors (Spearman, 1904). Let $x, y \in \mathbb{R}^m$ be two real-valued vectors. Let $R(x), R(y) \in \mathbb{R}^m$ denote their rank vectors, where $R(x)_k$ is the rank of $x_k$ among $\{x_\ell\}_{\ell=1}^m$ (ties resolved deterministically, e.g., by mid-ranks). Spearman's $\rho_S$ is

$$\rho_S(x, y) := \text{corr}\big(R(x), R(y)\big) = \frac{\sum_{k=1}^m \big(R(x)_k - \overline{R(x)}\big)\big(R(y)_k - \overline{R(y)}\big)}{\sqrt{\sum_{k=1}^m \big(R(x)_k - \overline{R(x)}\big)^2}\sqrt{\sum_{k=1}^m \big(R(y)_k - \overline{R(y)}\big)^2}} \in [-1, 1], \qquad (1)$$

where $\overline{R(x)} = \frac{1}{m}\sum_{k=1}^m R(x)_k$ and $\overline{R(y)} = \frac{1}{m}\sum_{k=1}^m R(y)_k$.

**Topological event ordering in $0$D and its link to MST.** In the 0D Vietoris–Rips filtration induced by $w$, connected components merge exactly when an edge $(i, j)$ with $w_{ij} \leq \alpha$ first connects two previously disconnected components. A classical equivalence (single-linkage clustering / Kruskal's algorithm) states that these $n-1$ merge events occur at the $n-1$ edge weights of an MST of $w$, sorted in nondecreasing order. We fix a deterministic tie-breaking rule so the MST is well-defined. Moreover, for any pair $(i, j)$, their merge time equals the smallest threshold at which they become connected, which can be read off from the MST as

$$m_w(i, j) := \min\{\alpha : i \text{ and } j \text{ are connected at threshold } \alpha\} = \max_{e \in \text{path}_{T_w}(i,j)} w_e. \qquad (2)$$

**Core pairs and NTS.** Let $E_w$ and $E_{\tilde{w}}$ be the MST edge sets under $w$ and $\tilde{w}$, and define $\mathcal{C} := E_w \cup E_{\tilde{w}}$. Indexing by $e = (i, j) \in \mathcal{C}$, we form aligned vectors and define NTS via Spearman correlation equation 1:

- **NTS-E:** $x_e := w_{ij}$, $\tilde{x}_e := \tilde{w}_{ij}$, $\quad$ NTS-E$(w, \tilde{w}) := \rho_S\big((x_e)_{e \in \mathcal{C}}, (\tilde{x}_e)_{e \in \mathcal{C}}\big)$.

- **NTS-M:** $x_e := m_w(i,j)$, $\tilde{x}_e := m_{\tilde{w}}(i,j)$, $\quad$ NTS-M$(w, \tilde{w}) := \rho_S\big((x_e)_{e \in \mathcal{C}}, (\tilde{x}_e)_{e \in \mathcal{C}}\big)$.

### 4.3 Formal Definition and Properties

Algorithms 1–2 summarize the computation of NTS-M and NTS-E. All MSTs are computed with a fixed deterministic tie-breaking rule, and ranks in Spearman's $\rho_S$ (Eq. 1) are computed with a fixed tie-handling convention (e.g., mid-ranks).

| **Algorithm 1:** NTS-M (merge-time based) | **Algorithm 2:** NTS-E (edge-distance based) |
|---|---|
| **Input:** Dissimilarity matrices $w, \tilde{w}$ | **Input:** Dissimilarity matrices $w, \tilde{w}$ |
| **Output:** NTS-M$(w, \tilde{w})$ | **Output:** NTS-E$(w, \tilde{w})$ |
| 1 $E_w \leftarrow$ edge set of MST$(w)$ | 1 $E_w \leftarrow$ edge set of MST$(w)$ |
| 2 $E_{\tilde{w}} \leftarrow$ edge set of MST$(\tilde{w})$ | 2 $E_{\tilde{w}} \leftarrow$ edge set of MST$(\tilde{w})$ |
| 3 $\mathcal{C} \leftarrow E_w \cup E_{\tilde{w}}$ | 3 $\mathcal{C} \leftarrow E_w \cup E_{\tilde{w}}$ |
| 4 $V \leftarrow \big(m_w(i,j)\big)_{(i,j) \in \mathcal{C}}$      // Eq. 2 | 4 $V \leftarrow \big(w_{ij}\big)_{(i,j) \in \mathcal{C}}$ |
| 5 $\tilde{V} \leftarrow \big(m_{\tilde{w}}(i,j)\big)_{(i,j) \in \mathcal{C}}$ | 5 $\tilde{V} \leftarrow \big(\tilde{w}_{ij}\big)_{(i,j) \in \mathcal{C}}$ |
| 6 **return** $\rho_S(V, \tilde{V})$      // Eq. 1 | 6 **return** $\rho_S(V, \tilde{V})$      // Eq. 1 |

We next state two basic properties. We use "weak order" to include ties: two vectors induce the same weak order on $\mathcal{C}$ if for all $e_1, e_2 \in \mathcal{C}$, $V_{e_1} < V_{e_2} \Leftrightarrow \tilde{V}_{e_1} < \tilde{V}_{e_2}$ and $V_{e_1} = V_{e_2} \Leftrightarrow \tilde{V}_{e_1} = \tilde{V}_{e_2}$.

**Theorem 4.1.** *Assume the rank vectors in Eq. 1 have nonzero variance. Then* NTS-M$(w, \tilde{w}) = 1$ *if and only if the merge-time values* $\{m_w(i,j)\}_{(i,j) \in \mathcal{C}}$ *and* $\{m_{\tilde{w}}(i,j)\}_{(i,j) \in \mathcal{C}}$ *induce the same weak order on* $\mathcal{C}$.

**Theorem 4.2.** *Assume the rank vectors in Eq. 1 have nonzero variance. If* NTS-E$(w, \tilde{w}) = 1$*, then* NTS-M$(w, \tilde{w}) = 1$*. The converse does not necessarily hold.*

Intuitively, NTS-E is stricter because it matches the rank structure of the underlying edge weights on $\mathcal{C}$, whereas NTS-M only requires agreement in the induced merge-event ordering.

## 5 Experiments

### 5.1 Analysis of Hierarchical Clustering Structures

We evaluate the two complementary components of our unified topological toolkit on controlled hierarchical-structure shifts. First, we use SRTD and SRTD-lite as *diagnostic* divergences: their cross-barcodes provide fine-grained, interpretable evidence of where paired representations differ (and, when desired, the associated divergence can also serve as a loss term). Second, we study NTS as a *normalized* topological similarity for robust cross-scenario comparison, and later illustrate how this topological perspective complements geometric baselines in CNN and LLM case studies.

**Clusters Experiment.** We test sensitivity to increasing structural dissimilarity by comparing a single cluster of 300 2D Gaussian points against variants where the points are partitioned into $k = 2, \ldots, 12$ clusters arranged on a circle. The results in Figure 4 reveal a clear performance divide: our proposed NTS and SRTD families correctly capture the expected trend of increasing dissimilarity. In contrast, CKA is largely insensitive to these structural changes, while RTD-lite produces an anomalous, inverted trend, confirming that the $\max(w, \tilde{w})$ component is essential for a robust divergence measure.

**UMAP Embeddings Experiment.** We test sensitivity to structural changes by generating a sequence of 2D UMAP embeddings (Damrich & Hamprecht, 2021) from the MNIST dataset (LeCun et al., 1998), varying the `n_neighbors` parameter to control the trade-off between local and global structure. Pairwise comparisons of these embeddings (Figure 5) demonstrate that our proposed methods, NTS and SRTD-lite,

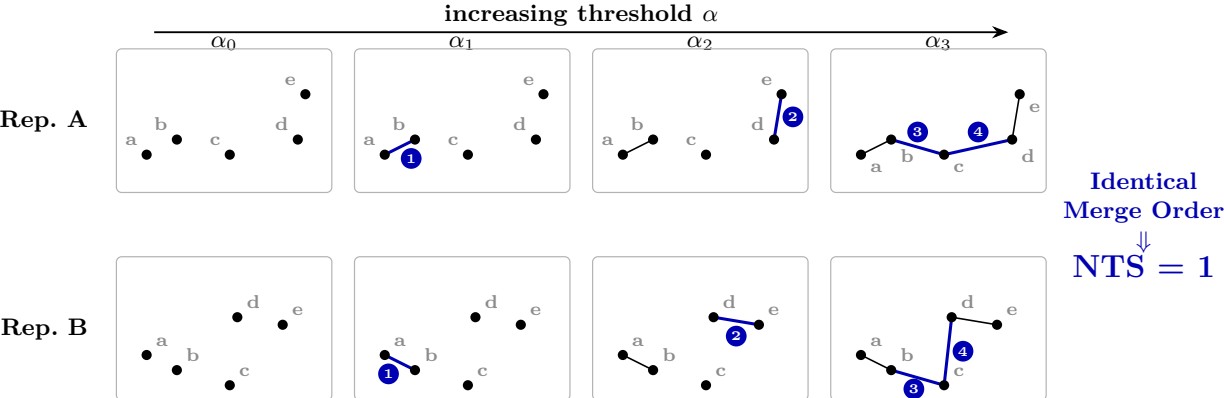

Figure 3: While Rep. A and Rep. B have distinct geometric layouts (CKA ≈ 0.68), their 0D merge events follow the exact same sequence (labeled ①–④). NTS yields a score of 1, reflecting topological consistency despite geometric variance.

track these changes with a smooth, monotonic response. In contrast, the CKA baseline fails to capture this gradual evolution, highlighting the superior sensitivity of our topological measures.

## 5.2 Efficiency as an Optimization Loss

We evaluate the practical utility of our divergence measures as loss terms for training an autoencoder, a task for which they are naturally suited. In this experiment, an autoencoder is trained to reduce the dimensionality of the F-MNIST and COIL-20 datasets to 16 (Xiao et al., 2017; Nene et al., 1996). It is crucial to note this is an **intra-family comparison**, designed to demonstrate that our proposed SRTD offers the best trade-off between performance and efficiency within the RTD class of methods. The results confirm that SRTD and SRTD-lite achieve top-tier performance on quality metrics while being faster than their predecessors. (Full results are provided in Appendix E).

## 5.3 Analyzing Structural Consistency and Functional Hierarchy

To evaluate our proposed methods in a practical deep learning context, we analyze the structural consistency of representations learned by an 8-layer `TinyCNN` (see Appendix C). Our experimental design on CIFAR-10 (Krizhevsky & Hinton, 2009) follows the evaluation protocols established in the foundational work by Kornblith et al. (2019). We extract representations from 5,000 test images across ten models trained with different random seeds.

The heatmaps in Figure 6, showing the average results over all 45 unique model pairs, provide two key insights into the behavior of these measures:

- **Normalized similarity landscapes.** We observe that NTS and CKA (as well as RSA, see Appendix G) exhibit a consistent, interpretable near-diagonal organization: representations are most similar to their immediate neighbors, with similarity decaying smoothly with depth distance. This confirms that these metrics correctly capture the hierarchical evolution of features. In stark contrast, the RTD family (represented by SRTD-lite, see Appendix D for detail) fails to recover this monotonic trend. Instead, it produces irregular heatmaps containing counter-intuitive inversions—for instance, an early layer (e.g., Layer 1) may appear structurally closer to the final layer (Layer 8) than a middle layer (Layer 4) does. These anomalies suggest that the unnormalized nature of RTD-style divergences, coupled with the heuristic nature of the 0.9-quantile normalization, makes them ill-suited for fine-grained cross-layer comparison, thereby highlighting the necessity of the rank-based NTS.

- **Complementary signal at functional transitions.** Beyond producing a coherent landscape, topology-aware measures (NTS and the RTD family, e.g., SRTD-lite) additionally highlight a sharp

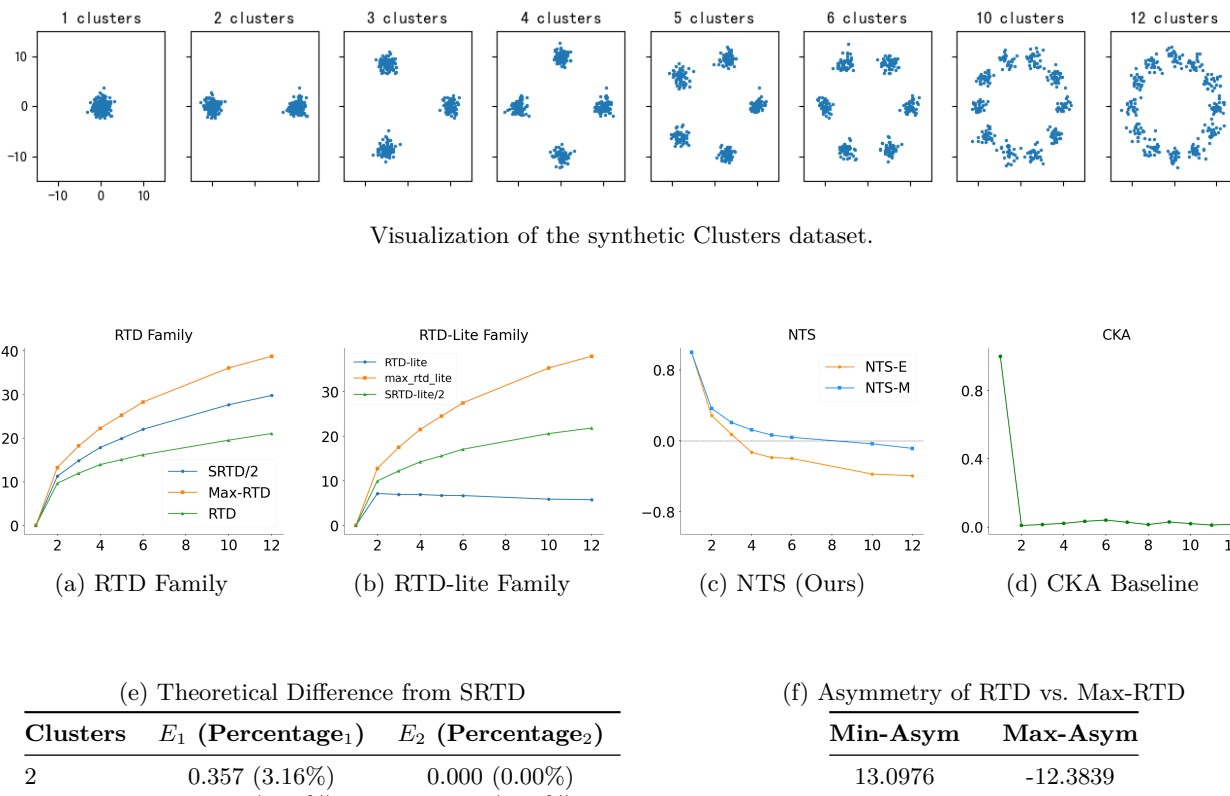

Visualization of the synthetic Clusters dataset.

| (a) RTD Family | (b) RTD-lite Family | (c) NTS (Ours) | (d) CKA Baseline |

(e) Theoretical Difference from SRTD

| Clusters | $E_1$ (**Percentage$_1$**) | $E_2$ (**Percentage$_2$**) |
|---|---|---|
| 2 | 0.357 (3.16%) | 0.000 (0.00%) |
| 3 | 0.493 (3.32%) | 0.013 (0.09%) |
| 4 | 0.441 (2.47%) | 0.061 (0.34%) |
| 5 | 0.451 (2.26%) | 0.039 (0.20%) |
| 6 | 0.347 (1.57%) | 0.060 (0.27%) |
| 10 | 0.263 (0.95%) | 0.043 (0.15%) |
| 12 | 0.226 (0.76%) | 0.046 (0.15%) |

(f) Asymmetry of RTD vs. Max-RTD

| Min-Asym | Max-Asym |
|---|---|
| 13.0976 | -12.3839 |
| 11.2554 | -10.2954 |
| 10.8131 | -10.0535 |
| 10.3320 | -9.5084 |
| 9.4315 | -8.8572 |
| 8.3074 | -7.8674 |
| 7.6888 | -7.3296 |

Figure 4: **Comprehensive analysis of the RTD framework on the synthetic Clusters dataset. (a–d)** Performance comparison across different measures; note the superior sensitivity of NTS and SRTD families compared to CKA and RTD-lite. **(e)** Evaluation of the small theoretical gap between SRTD and symmetrized directional variants, where $E_1$ and $E_2$ quantify the contribution of private topological features unique to individual filtrations. **(f)** Visualization of the strong asymmetry and inherent complementarity between RTD and Max-RTD. *Definitions:* $E_1 = (\text{RTD}(w, \tilde{w}) + \text{Max-RTD}(w, \tilde{w}) - \text{SRTD})/2$; Min-Asym = $\text{RTD}(w, \tilde{w}) - \text{RTD}(\tilde{w}, w)$.

transition at the final pooling layer. This aligns with the architectural shift from local feature extraction to global aggregation, suggesting that topological tools can provide a complementary diagnostic cue for identifying structural changes that may not be obvious from magnitude-based similarity alone.

## 5.4 Analysis of Large Language Model Representations

Finally, we extend our evaluation to Large Language Models (LLMs) to assess the utility of NTS. Our goal is to illustrate how topological similarity measures can enrich the analysis provided by established geometric methods like CKA. We investigate whether a topological lens can uncover structural nuances—such as family-specific hierarchies—that may be less apparent under standard geometric analysis due to phenomena like distance saturation.

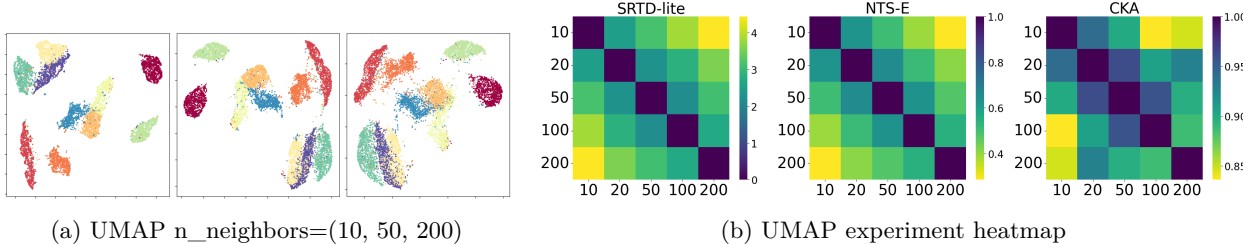

(a) UMAP n_neighbors=(10, 50, 200)   (b) UMAP experiment heatmap

Figure 5: UMAP experiment

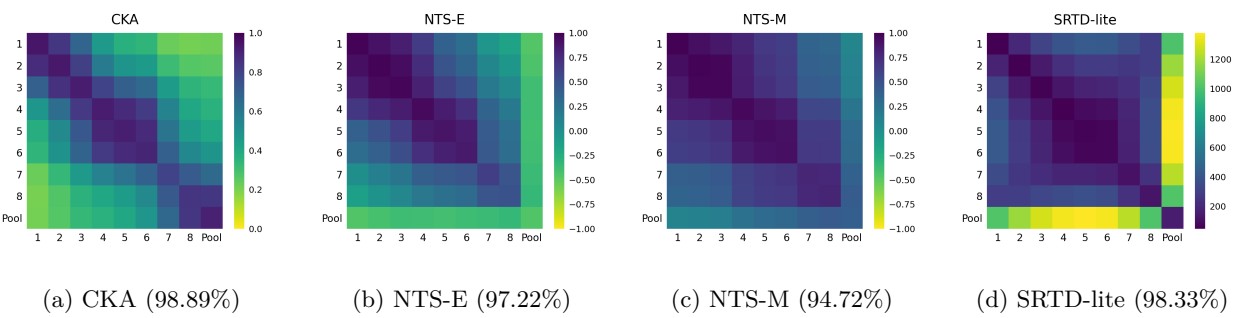

(a) CKA (98.89%)    (b) NTS-E (97.22%)    (c) NTS-M (94.72%)    (d) SRTD-lite (98.33%)

Figure 6: Average layer-wise similarity comparison over 45 pairs of trained `TinyCNN`s. While CKA (a) and NTS (b, c) both produce interpretable patterns within convolutional layers, only the topological measures (b–d) capture the sharp structural break at the final pooling layer—a functional shift missed by geometric analysis.

Our methodology is closely adapted from REEF (Zhang et al., 2024), a recent study that established a robust protocol for fingerprinting and comparing LLM representations. REEF identified that certain datasets are particularly effective at eliciting discriminative features that highlight inter-model differences. Following their findings, we conduct our analysis on two such datasets: TruthfulQA (Lin et al., 2021) and Toxi-Gen (Hartvigsen et al., 2022). For each dataset, we adopt the REEF protocol of extracting the last-token representation from every Transformer layer across 1,000 randomly sampled QA pairs.

**Identifying Intra-Model Hierarchical Patterns.** Our evaluation of intra-model layer similarity yields a compelling empirical finding regarding structural consistency. We observe that NTS uncovers highly consistent hierarchical "fingerprints" across models within the same family (Qwen, InternLM, Baichuan, and Llama). This aligns with the intuition that models sharing a common lineage should preserve their fundamental topological structure despite post-training refinements. In contrast, CKA does not consistently exhibit this family-wise regularity, as summarized in Figure 7. While it captures similar patterns within the InternLM family, it tends to diverge in others—often due to score saturation (e.g., Llama) or sensitivity to fine-tuning shifts (e.g., Qwen and Baichuan). This suggests that NTS provides a valuable, distinctive lens for characterizing the conserved functional hierarchy of LLMs, where geometric measures may be obscured by distance saturation.

**Inter-Model Similarity Analysis** Finally, we compare the ability of NTS and CKA to map the relationships between different LLM families. For this analysis, we focus on the last-token representation from the 6th Transformer layer, which empirically yielded the most discriminative features. Crucially, we apply Z-score normalization across the feature dimension before computing NTS to mitigate variance in individual activations. Detailed ablation studies on layer selection and normalization effects are provided in Appendix I.2.

Following the methodology of REEF (Zhang et al., 2024), we present the results on the TruthfulQA dataset in Figure 8. This visualization highlights a fundamental behavioral distinction between the two measures.

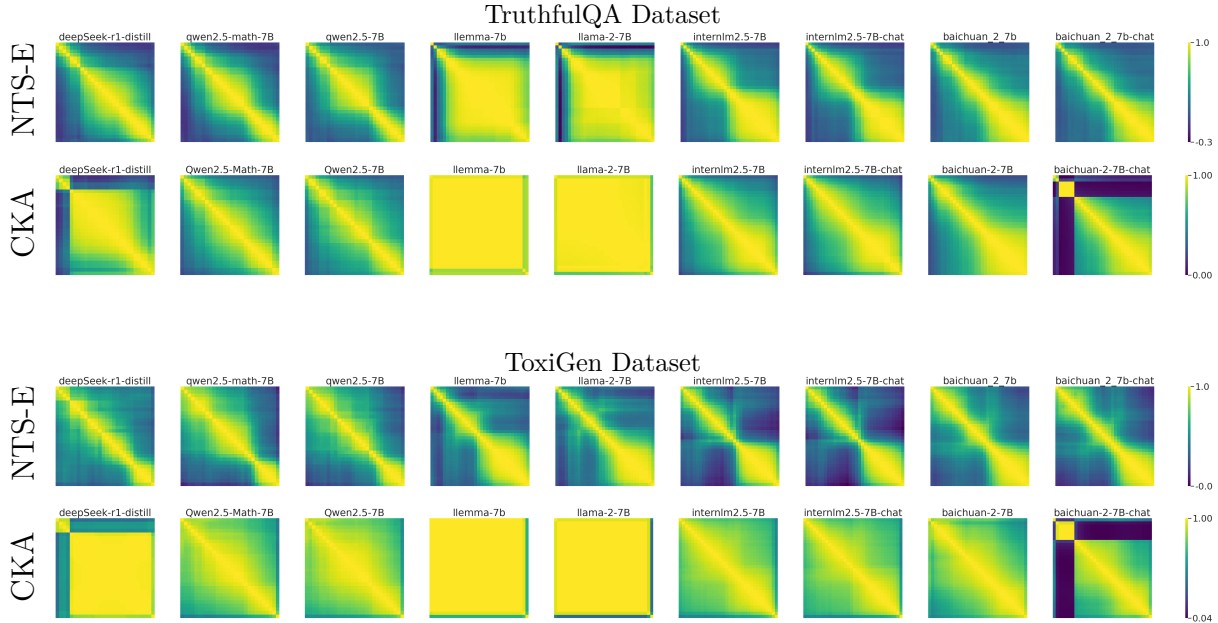

Figure 7: Intra-model layer similarity for LLM families on the TruthfulQA (top half) and ToxiGen (bottom half) datasets. NTS (top row of each pair) consistently reveals structured hierarchical patterns. In contrast, CKA (bottom row of each pair) often produces saturated or inconsistent heatmaps, failing on most families except InternLM.

While both metrics generally assign high scores to cross-family comparisons, CKA exhibits significant **score saturation**. As shown in Figure 8a, CKA scores for most non-Llama model pairs cluster near the maximum ($> 0.8$), severely limiting the ability to distinguish between distinct families such as Qwen, Mistral, and InternLM. In contrast, NTS scores (Figure 8b) are less saturated and more broadly distributed, offering a sharper, more discriminative view of the model landscape.

A particularly illuminating case involves `DeepSeek-R1-Ds` (Guo et al., 2025), a model distilled from `Qwen2.5-Math-7B` (Yang et al., 2024). Here, CKA yields a surprisingly low similarity score between the distilled model and its parent `Qwen2.5` family, failing to reflect their known lineage. Conversely, NTS-E successfully identifies a high structural similarity between them. This suggests that by focusing on topological merge orders rather than pure geometric alignment, NTS captures robust structural signals that persist even when geometric measures are disrupted by distillation-induced shifts.

Collectively, these findings underscore a key insight: while advanced training techniques like distillation may significantly alter the geometric layout of representation spaces—thereby obscuring relationships under CKA—the underlying topological backbone often remains preserved. NTS successfully captures this conserved structural heritage, validating its potential as a reliable tool for mapping the evolutionary genealogy of Large Language Models in an increasingly complex ecosystem.

## 6 Computational Efficiency and Scalability

Our proposed toolkit is designed for both scalability and analytical power. A formal complexity analysis shows that while the full SRTD is computationally intensive, the core components of our framework are highly efficient. Both SRTD-lite and NTS-E operate in $O(n^2(\alpha_{\mathrm{uf}}(n) + d))$ time, where $\alpha_{\mathrm{uf}}(n)$ denotes the inverse Ackermann factor from union-find operations. This cost is primarily dominated by the pairwise distance calculation and the Minimum Spanning Tree (MST) construction.

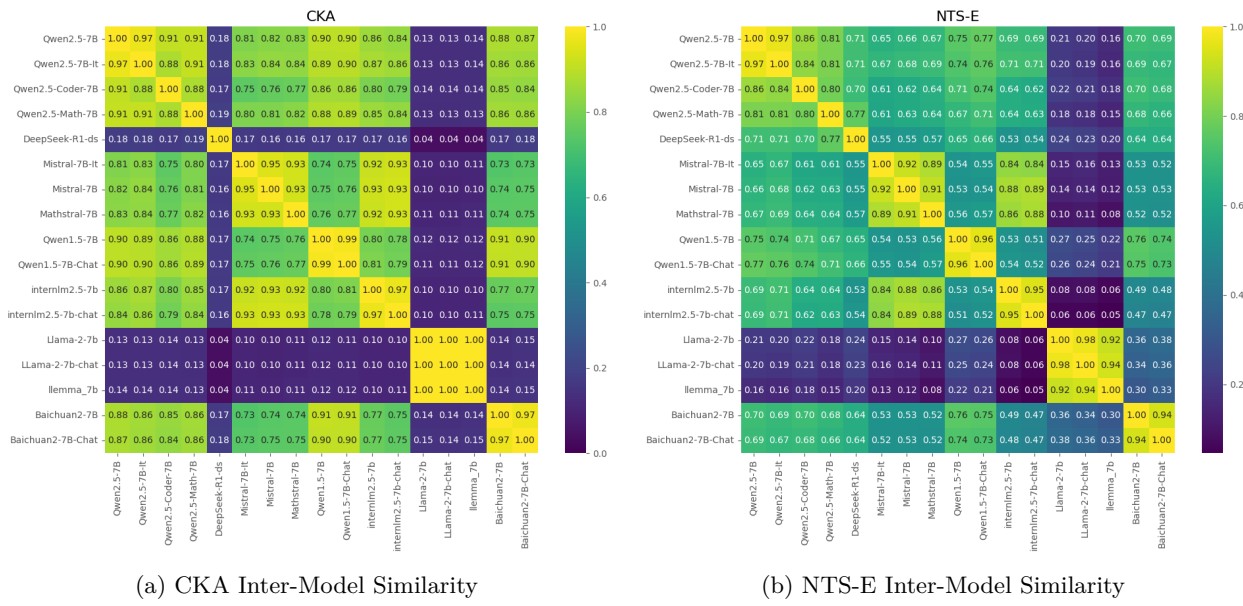

(a) CKA Inter-Model Similarity     (b) NTS-E Inter-Model Similarity

Figure 8: Inter-model similarity maps for 17 LLMs

To empirically validate this scalability, we conducted a runtime benchmark using representations from a TinyCNN trained on CIFAR-10. We varied the sample size $N$ from 5,000 to 30,000 and measured the end-to-end execution time. The results in Figure 9 unequivocally show that NTS-E exhibits the best scalability, followed by SRTD-lite, with RTD-lite being the slowest due to its triple MST calculation.

This significant efficiency gain in NTS-E stems from two key factors:

1. **No Normalization Required:** Being a rank-based measure, NTS-E operates directly on raw distance matrices, bypassing the costly quantile calculation and matrix division required by RTD and SRTD.

2. **Minimal Memory Footprint:** NTS-E avoids constructing dense auxiliary matrices (e.g., $\min(w, \tilde{w})$), reducing peak memory usage from $O(3N^2)$ to $O(2N^2)$, making it the most memory-efficient method.

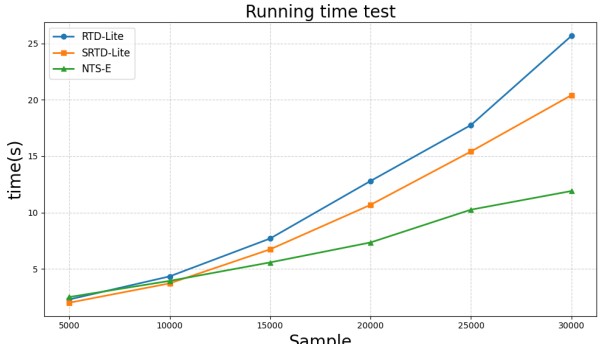

Figure 9: Runtime comparison on CIFAR-10 representations with varying sample sizes.

## 7 Conclusion

In summary, we introduce a complementary topological toolkit. These methods offer a powerful choice for representation analysis. While NTS is ideal for obtaining a single, stable similarity score, SRTD-lite offers in-depth diagnostics (Table 5) and can serve as an effective loss term. To achieve high scalability, our lightweight variants rely primarily on 0-dimensional features. While this trades off the unit-level, higher-order topological insights captured by methods like feature entropy (Zhao & Zhang, 2022), it enables highly efficient macro-level benchmarking. Another limitation is that NTS, in its current form, is an analysis-only measure. Its non-differentiable nature prevents its use in direct model optimization. Therefore, a crucial avenue for future research is to develop a differentiable formulation of NTS, enabling it to guide representation learning.

## Broader Impact Statement

This work provides foundational theoretical tools for understanding neural network representations. By advancing representation analysis, it indirectly benefits model auditing and interpretability. As an abstract methodological study, it does not directly introduce negative societal impacts or dual-use risks.

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

# A  Definition and Algorithm

**Definition A.1** (Max-RTD)**.** For two point clouds $P$ and $P'$ with a one-to-one correspondence, the distance matrix of their auxiliary graph $\hat{\mathcal{G}}'_{max}$ is given by $M_{max}$ (Matrix 1c). The sum of the lengths of the persistent homology barcodes of $\hat{\mathcal{G}}'_{max}$ is defined as $Max\text{-}RTD(w, \tilde{w})$. Its chain complex is homotopy equivalent to the mapping cone of the inclusion map $f' : C_*(R_\alpha(\mathcal{G}^w) \cap R_\alpha(\mathcal{G}^{\tilde{w}})) \to C_*(R_\alpha(\mathcal{G}^w))$.

## A.1  SRTD Algorithm

---

**Algorithm 3:** Symmetric Representation Topology Divergence (SRTD) Calculation

---

**Input:** Pairwise distance matrices $w, \tilde{w}$
**Output:** A set of divergence scores $\{SRTD_i\}_{i \geq 0}$ for each dimension $i$
**1** $w_{norm}, \tilde{w}_{norm} \leftarrow$ Normalize $w, \tilde{w}$ by their 0.9 quantiles;
**2** $w_{min} \leftarrow \min(w_{norm}, \tilde{w}_{norm})$;
**3** $w_{max} \leftarrow \max(w_{norm}, \tilde{w}_{norm})$;
**4** Construct the symmetric auxiliary matrix $M_{sym}$ using $w_{min}$ and $w_{max}$ (see Matrix 1a);
**5** **for** *each dimension of interest* $i \in \{0, 1, \dots\}$ **do**
**6**     Compute barcodes: $B_i \leftarrow \text{PersistentHomology}(M_{sym}, i)$;
**7**     Compute divergence score: $SRTD_i \leftarrow \sum_{(b,d) \in B_i}(d - b)$;
**8** **end**
**9** **return** $\{SRTD_i\}_{i \geq 0}$;

---

## A.2  SRTD-lite Barcode Algorithm

---

**Algorithm 4:** Computation of SRTD-lite Barcode

---

**Input:** Weight matrices $D_1, D_2$
**Output:** A multiset of intervals (the SRTD-L-Barcode)
**1** **procedure** SRTD-L-Barcode($D_1, D_2$)
**2**     $D'_1, D'_2 \leftarrow$ Normalize $D_1, D_2$ by their 0.9 quantiles;
**3**     $D_{\min} \leftarrow$ Element-wise minimum of $D'_1$ and $D'_2$;
**4**     $D_{\max} \leftarrow$ Element-wise maximum of $D'_1$ and $D'_2$;
**5**     $E_{\min} \leftarrow \text{Sort}(\text{MST}(D_{\min}))$;
**6**     $E_{\max} \leftarrow \text{Sort}(\text{MST}(D_{\max}))$;
**7**     $BarcodeSet \leftarrow [\,]$;
**8**     $SubTree \leftarrow$ Empty graph with $N$ vertices;
**9**     **foreach** *edge* $e = (u, v)$ *with weight* $w_{birth}$ *in* $E_{\min}$ **do**
**10**         **if** *u and v are not connected in SubTree* **then**
**11**             $TemporaryGraph \leftarrow \text{copy}(SubTree)$;
**12**             **foreach** *edge* $e' = (u', v')$ *with weight* $w_{death}$ *in* $E_{\max}$ **do**
**13**                 Add $e'$ to $TemporaryGraph$;
**14**                 **if** *u and v are connected in TemporaryGraph* **then**
**15**                     Add $(w_{birth}, w_{death})$ to $BarcodeSet$;
**16**                     ;
**17**                 **end**
**18**             **end**
**19**             Add $e$ to $SubTree$;
**20**         **end**
**21**     **end**
**22**     **return** $BarcodeSet$;

---

## B  Proofs

### B.1  Statement in Definition

We first prove the following lemmas, which are stated in Definition A.1 and Definition 3.1. The construction and proof for this part refer to Barannikov et al. (2021a). Let $A = R_\alpha(\mathcal{G}^w)$ and $B = R_\alpha(\mathcal{G}^{\tilde{w}})$:

**Lemma B.1.** *There exists a specially constructed auxiliary graph $\hat{\mathcal{G}}'_{max}$ such that its chain complex is homotopy equivalent to the mapping cone $\mathrm{Cone}(f')$, where $f' : C_*(A \cap B) \to C_*(A)$ is a chain map induced by the inclusion.*

$$R_\alpha(\hat{\mathcal{G}}'_{max}) \sim \mathrm{Cone}\left(R_\alpha(\mathcal{G}^{\max(w,\tilde{w})}) \to R_\alpha(\mathcal{G}^w)\right)$$

**Lemma B.2.** *Similarly, there exists a specially constructed auxiliary graph $\hat{\mathcal{G}}'_{sym}$ such that its chain complex is homotopy equivalent to the mapping cone $\mathrm{Cone}(f')$, where $f' : C_*(A \cap B) \to C_*(A \cup B)$ is a chain map induced by the inclusion.*

$$R_\alpha(\hat{\mathcal{G}}'_{sym}) \sim \mathrm{Cone}\left(R_\alpha(\mathcal{G}^{\max(w,\tilde{w})}) \to R_\alpha(\mathcal{G}^{\min(w,\tilde{w})})\right)$$

*Proof.* The mapping cone we are interested in is constructed from the direct sum of the following chain complexes:

$$\mathrm{Cone}(f') = C_*(A \cap B)[-1] \oplus C_*(A)$$

Following the construction from the RTD paper, we can propose two auxiliary graph schemes: The vertex set of the auxiliary graph $\hat{\mathcal{G}}'_{max}$ is composed of the original vertices $v'_i$, mirrored vertices $v_i$, and a special vertex $O$. Its distance rules are defined as follows: $d'_{v_i v_j} = \max(w_{ij}, \tilde{w}_{ij}), d'_{v'_i v'_j} = w_{ij}, d'_{v_i v'_i} = 0, d'_{Ov_i} = 0,$ $d'_{Ov'_i} = +\infty, d'_{v_i v'_j} = \max(w_{ij}, \tilde{w}_{ij})$

The vertex set of the auxiliary graph $\hat{\mathcal{G}}'_{sym}$ is composed of twice the number of original vertices and $O$. $d'_{v_i v_j} = \max(w_{ij}, \tilde{w}_{ij}), d'_{v'_i v'_j} = \min(w_{ij}, \tilde{w}_{ij}), d'_{v_i v'_i} = 0, d'_{Ov_i} = 0, d'_{Ov'_i} = +\infty, d'_{v_i v'_j} = \max(w_{ij}, \tilde{w}_{ij})$

For the auxiliary graph $R_\alpha(\hat{\mathcal{G}}'_{max})$, there are three types of simplices:

- $A_{i_1} \ldots A_{i_k} A'_{i_k} \ldots A'_{i_n}$, where $\max(w_{A_{i_r} A_{i_s}}, \tilde{w}_{A_{i_r} A_{i_s}}) \leq \alpha$ for $r \leq k$, and $w_{A_{i_r} A_{i_s}} \leq \alpha$ for $r, s \geq k$.

- $A_{i_1} \ldots A_{i_k} A'_{i_{k+1}} \ldots A'_{i_n}$, where $\max(w_{A_{i_r} A_{i_s}}, \tilde{w}_{A_{i_r} A_{i_s}}) \leq \alpha$ for $r \leq k$, and $w_{A_{i_r} A_{i_s}} \leq \alpha$ for $r, s \geq k+1$.

- $O A_{i_1} A_{i_2} \ldots A_{i_n}$, where $\max(w_{A_{i_r} A_{i_s}}, \tilde{w}_{A_{i_r} A_{i_s}}) \leq \alpha$.

**Forward Map**

$$\psi' : \mathrm{Cone}(f') \to R_\alpha(\hat{\mathcal{G}}'_{max})$$

- For $c \in C_*(A \cap B)[-1]$ (of the form $A_{i_1} \ldots A_{i_n}[-1]$):

$$\psi'(c) = O A_{i_1} \ldots A_{i_n} + \sum_{k=1}^{n} A_{i_1} \ldots A_{i_k} A'_{i_k} \ldots A'_{i_n}$$

- For $a \in C_*(A)$ (of the form $A_{i_1} \ldots A_{i_n}$):

$$\psi'(a) = A'_{i_1} \ldots A'_{i_n}$$

**Backward Map**

$$\tilde{\psi}' : R_\alpha(\hat{\mathcal{G}}'_{max}) \to \mathrm{Cone}(f')$$

- $\tilde{\psi}'(O A_{i_1} \ldots A_{i_n}) = A_{i_1} \ldots A_{i_n}[-1]$

- $\tilde{\psi}'(A'_{i_1} \ldots A'_{i_n}) = A_{i_1} \ldots A_{i_n}$

- $\tilde{\psi}'(\Delta) = 0$ (for all other types of simplices $\Delta$)

**Homotopy Operator H** For the second type of simplex:

$$H : A_{i_1} \ldots A_{i_k} A'_{i_{k+1}} \ldots A'_{i_n} \to \sum_{l=1}^{k} A_{i_1} \ldots A_{i_l} A'_{i_l} \ldots A'_{i_n}, 1 \le k \le n$$

For all other simplices:

$$H(\Delta) = 0$$

Therefore, $\tilde{\psi}' \circ \psi' = \mathrm{Id}$ and $\psi' \circ \tilde{\psi}' - \mathrm{Id} = H\partial - \partial H$. This proves B.1, and B.2 can be proven similarly. $\square$

## B.2 Proof of Theorem 3.3

Let's prove Theorem 3.3. To prove the theorem, we just need to prove the following lemma:

**Lemma B.3.** *For any dimension $i$, the Betti numbers of the three auxiliary graphs satisfy the following relation:*

$$\beta_i^{\min}(\alpha) + \beta_i^{\max}(\alpha) - \beta_i^{sym}(\alpha) = \dim(ker(\gamma_i)) + \dim(ker(\gamma_{i-1}))$$

*Proof.* We have the following inclusion of simplicial complexes:

$$R_\alpha(\mathcal{G}^{\max(w,\tilde{w})}) \subseteq R_\alpha(\mathcal{G}^w) \subseteq R_\alpha(\mathcal{G}^{\min(w,\tilde{w})})$$

This forms a triple of complexes, which gives rise to a standard short exact sequence of their chain complexes:

$$0 \to C_*(R_\alpha(\mathcal{G}^w), R_\alpha(\mathcal{G}^{\max(w,\tilde{w})})) \to C_*(R_\alpha(\mathcal{G}^{\min(w,\tilde{w})}), R_\alpha(\mathcal{G}^{\max(w,\tilde{w})})) \to C_*(R_\alpha(\mathcal{G}^{\min(w,\tilde{w})}), R_\alpha(\mathcal{G}^w)) \to 0$$

This, in turn, induces the following long exact sequence in homology:

$$\cdots \to H_n(R_\alpha(\mathcal{G}^w), R_\alpha(\mathcal{G}^{\max(w,\tilde{w})})) \to H_n(R_\alpha(\mathcal{G}^{\min(w,\tilde{w})}), R_\alpha(\mathcal{G}^{\max(w,\tilde{w})}))$$

$$\to H_n(R_\alpha(\mathcal{G}^{\min(w,\tilde{w})}), R_\alpha(\mathcal{G}^w)) \xrightarrow{\partial_*} H_{n-1}(R_\alpha(\mathcal{G}^w), R_\alpha(\mathcal{G}^{\max(w,\tilde{w})})) \to \cdots$$

Since the relative homology groups are isomorphic to the homology groups of the corresponding mapping cones, we have the following long exact sequence for the auxiliary graphs:

$$\cdots \to H_i(R_\alpha(\hat{\mathcal{G}}'_{max})) \xrightarrow{\gamma_i} H_i(R_\alpha(\hat{\mathcal{G}}'_{sym})) \xrightarrow{\beta_i} H_i(R_\alpha(\hat{\mathcal{G}}'_{min})) \xrightarrow{\delta_i} H_{i-1}(R_\alpha(\hat{\mathcal{G}}'_{max})) \to \cdots$$

where $\gamma_i, \beta_i, \delta_i$ are the homomorphism maps in the sequence. For any segment of an exact sequence of vector spaces $U \xrightarrow{f} V \xrightarrow{g} W$, we have $\mathrm{im}(f) = \ker(g)$. By the rank-nullity theorem, $\dim(V) = \dim(\ker(g)) + \dim(\mathrm{im}(g))$. Substituting $\mathrm{im}(f) = \ker(g)$, we get $\dim(V) = \dim(\mathrm{im}(f)) + \dim(\mathrm{im}(g))$. Therefore, the dimensions of the homology groups of the auxiliary graphs (i.e., the Betti numbers $\beta_i(\alpha)$) can be expressed as:

$$\beta_i^{\max}(\alpha) = \dim(H_i(R_\alpha(\hat{\mathcal{G}}'_{max}))) = \dim(\mathrm{im}(\delta_{i+1})) + \dim(\mathrm{im}(\gamma_i)) \tag{3}$$

$$\beta_i^{\mathrm{sym}}(\alpha) = \dim(H_i(R_\alpha(\hat{\mathcal{G}}'_{sym}))) = \dim(\mathrm{im}(\gamma_i)) + \dim(\mathrm{im}(\beta_i)) \tag{4}$$

$$\beta_i^{\min}(\alpha) = \dim(H_i(R_\alpha(\hat{\mathcal{G}}'_{min}))) = \dim(\mathrm{im}(\beta_i)) + \dim(\mathrm{im}(\delta_i)) \tag{5}$$

By substituting equation 3, equation 4, and equation 5, we obtain:

$$\beta_i^{\min}(\alpha) + \beta_i^{\max}(\alpha) - \beta_i^{\mathrm{sym}}(\alpha)$$
$$= \big(\dim(\mathrm{im}(\beta_i)) + \dim(\mathrm{im}(\delta_i))\big)$$
$$\quad + \big(\dim(\mathrm{im}(\delta_{i+1})) + \dim(\mathrm{im}(\gamma_i))\big)$$
$$\quad - \big(\dim(\mathrm{im}(\gamma_i)) + \dim(\mathrm{im}(\beta_i))\big)$$
$$= \dim(\mathrm{im}(\delta_{i+1})) + \dim(\mathrm{im}(\delta_i))$$
$$= \dim(\ker(\gamma_i)) + \dim(\ker(\gamma_{i-1}))$$

By integrating both sides of Lemma B.3 with respect to filtration radius $\alpha$, we obtain its conclusion. This completes the proof of Lemma B.3 and Theorem 3.3. □

## B.3 Proof of Corollary

**Proof of Corollary 3.4** From definition, we have

$$RTD\text{-}lite(P, P') = \frac{(mst(\mathcal{G}^w) - mst(\mathcal{G}^{\min(w,\tilde{w})})) + (mst(\mathcal{G}^{\tilde{w}}) - mst(\mathcal{G}^{\min(w,\tilde{w})}))}{2}$$

$$Max\text{-}RTD\text{-}lite(P, P') = \frac{(mst(\mathcal{G}^{\max(w,\tilde{w})}) - mst(\mathcal{G}^w)) + (mst(\mathcal{G}^{\max(w,\tilde{w})}) - mst(\mathcal{G}^{\tilde{w}}))}{2}$$

$$SRTD\text{-}lite(P, P') = mst(\mathcal{G}^{\max(w,\tilde{w})}) - mst(\mathcal{G}^{\min(w,\tilde{w})})$$

Summing the three equations above completes the proof.

**Proof of Corollary 3.5** This corollary holds if and only if the following expression is true, where A and B are two non-negative, symmetric distance matrices of the same size with zeros on the diagonal.

*Proof.*

$$\text{MST}(\max(A, B)) + \text{MST}(\min(A, B)) \geq \text{MST}(A) + \text{MST}(B). \qquad (\star)$$

Let the graph have $n$ vertices and an edge set $E$. We can view a weight matrix $W$ as a function that assigns a non-negative weight $W_e$ to each edge $e \in E$. For any non-negative weight matrix $W$, let $E_{\leq t}(W) := \{e \in E : W_e \leq t\}$ be the set of edges with weight at most $t$, and let $\kappa_W(t)$ be the number of connected components in the graph $(V, E_{\leq t}(W))$. A standard result from Kruskal's algorithm gives the MST weight as an integral:

$$\text{MST}(W) = \int_0^\infty \big(\kappa_W(t) - 1\big)\, dt. \qquad (6)$$

The element-wise min and max operations on weight matrices correspond to the union and intersection of their threshold edge sets:

$$E_{\leq t}(\max(A, B)) = E_{\leq t}(A) \cap E_{\leq t}(B), \qquad (7)$$
$$E_{\leq t}(\min(A, B)) = E_{\leq t}(A) \cup E_{\leq t}(B).$$

Let $\kappa(S)$ be the number of connected components of the graph induced by an edge set $S \subseteq E$. A fundamental result in graph theory and matroid theory is that the rank function $r(S) = n - \kappa(S)$ is submodular. Consequently, $\kappa(S)$ is supermodular:

$$\kappa(X \cap Y) + \kappa(X \cup Y) \geq \kappa(X) + \kappa(Y), \quad \forall X, Y \subseteq E. \qquad (8)$$

Substituting equation 7 into equation 8 with $X = E_{\leq t}(A)$ and $Y = E_{\leq t}(B)$, we get for every $t \geq 0$:

$$\kappa_{\max(A,B)}(t) + \kappa_{\min(A,B)}(t) \geq \kappa_A(t) + \kappa_B(t).$$

Integrating over $t \in [0, \infty)$, and applying the formula equation 6 yields the desired inequality $(\star)$. □

## B.4 Proofs for NTS Theorems

### B.4.1 Proof of Theorem 4.1

*Proof.* By definition, $NTS\text{-}M(P, P')$ is the Spearman's rank correlation coefficient, $\rho$, between the merge-time vectors $T$ and $\tilde{T}$. Let $R = \text{rank}(T)$ and $\tilde{R} = \text{rank}(\tilde{T})$ be the rank vectors computed with the *same deterministic tie-handling rule* (e.g., mid-ranks) on both sides. Recall that Spearman's $\rho$ is the Pearson's correlation applied to these ranks: $\rho = \text{corr}(R, \tilde{R})$.

**corr**$= 1 \implies$ **Identical Rank Weak Order**  We assume the non-degenerate case where $|E_{core}| \geq 2$ and both rank vectors have nonzero variance (i.e., not all merge times are identical). In this case, the Pearson correlation $\text{corr}(R, \tilde{R}) = 1$ if and only if there exist constants $a \in \mathbb{R}$ and $b > 0$ such that $\tilde{R} = a + bR$ holds entrywise. Since $b > 0$, this linear relationship ensures that the weak order of the ranks is identical. That is, for any two core pairs $e_1, e_2$:

$$R(e_1) < R(e_2) \iff \tilde{R}(e_1) < \tilde{R}(e_2),$$
$$R(e_1) = R(e_2) \iff \tilde{R}(e_1) = \tilde{R}(e_2).$$

**Identical Rank Weak Order** $\iff$ **Identical Merge-Time Weak Order**  Under a fixed tie-handling rule, the rank function is order-preserving and tie-preserving, and therefore also order-reflecting. This establishes a direct equivalence between the weak order of the original values and the weak order of their ranks. Thus, for any $e_1, e_2$:

$$T(e_1) < T(e_2) \iff R(e_1) < R(e_2),$$
$$T(e_1) = T(e_2) \iff R(e_1) = R(e_2).$$

The same equivalence holds for $\tilde{T}$ and $\tilde{R}$.

**Conclusion**  Chaining the equivalences from Step 1 and Step 2, we conclude that $NTS\text{-}M(P, P') = 1$ is equivalent to the statement that the merge-time weak order is identical.

To explicitly prove the biconditional ("if and only if") nature:

($\Rightarrow$) If $NTS\text{-}M = 1$, Step 1 shows the rank weak order is identical, which by Step 2 implies the merge-time weak order is identical.

($\Leftarrow$) Conversely, if the merge-time weak order is identical, then by Step 2, the rank weak order must be identical. This implies that the rank vectors themselves are identical, $R = \tilde{R}$. In the non-degenerate case, the correlation of a vector with itself is 1, so $\rho = \text{corr}(R, \tilde{R}) = 1$.

Therefore, $NTS\text{-}M(P, P') = 1$ if and only if the merge-time weak orders coincide. $\qquad\square$

### B.4.2  Proof of Theorem 4.2

*Proof.* The proof consists of two parts.

$NTS\text{-}E = 1 \implies NTS\text{-}M = 1$  Assume the non-degenerate case where $|E_{core}| \geq 2$ and the rank vectors of the edge distances have nonzero variance. The premise is $NTS\text{-}E(P, P') = 1$. By Theorem 4.1, this is equivalent to the statement that the weak order of the edge distances coincides for all core edges $e \in E_{core}$.

All MST and merge-time computations are performed on the fixed core graph $G_{core} = (V, E_{core})$, using the same deterministic tie-handling (e.g., mid-ranks) and tie-breaking (e.g., by edge index) rules on both sides.

The coincidence of the weak order of weights $\{w_e\}_{e \in E_{core}}$ and $\{\tilde{w}_e\}_{e \in E_{core}}$ implies that there exists a strictly increasing map $g$ defined on the finite set of values taken by $w$ on $E_{core}$, such that $\tilde{w}_e = g(w_e)$ for all $e \in E_{core}$. Because $g$ is strictly increasing, it does not change the sorted order of edges processed by Kruskal's algorithm on $G_{core}$. Therefore, the sequence of component merges is identical for both $w$ and $\tilde{w}$, and the resulting MSTs are identical. Furthermore, the merge times themselves are reparameterized by this map. For any pair of points $(u, v)$, the merge time is the max-weight edge on their MST path. Thus, for any core edge $e$:

$$T(e) = \max_{e' \in \text{path}(e)} w_{e'} \implies \tilde{T}(e) = \max_{e' \in \text{path}(e)} \tilde{w}_{e'} = \max_{e' \in \text{path}(e)} g(w_{e'}) = g\left(\max_{e' \in \text{path}(e)} w_{e'}\right) = g(T(e))$$

Since $\tilde{T}(e) = g(T(e))$ for a strictly increasing function $g$, the weak order of the merge times is preserved. By Theorem 4.1, this implies $NTS\text{-}M(P, P') = 1$.

**The Converse is Not Necessarily True** To prove the converse is false, we provide a minimal, reproducible counterexample where $NTS\text{-}M = 1$ but $NTS\text{-}E < 1$. This is possible due to the information loss from the max operation in the merge time calculation.

Let the set of vertices be $V = \{1, 2, 3, 4\}$ and the set of core edges be $E_{core} = \{(1, 2), (2, 3), (3, 4), (1, 3), (2, 4)\}$. Consider two weight functions $w$ and $\tilde{w}$ on $E_{core}$:

- $w$: $w_{12} = 2, w_{23} = 8, w_{34} = 10, w_{13} = 9, w_{24} = 7$.

- $\tilde{w}$: $\tilde{w}_{12} = 9, \tilde{w}_{23} = 7, \tilde{w}_{34} = 10, \tilde{w}_{13} = 8, \tilde{w}_{24} = 2$.

1. **NTS-E Score:** The vector of weights for $w$ on $E_{core}$ (ordered lexicographically) is $(2, 9, 7, 8, 10)$, which has a rank vector of $(1, 4, 2, 3, 5)$. The vector for $\tilde{w}$ is $(9, 8, 2, 7, 10)$, with a rank vector of $(4, 3, 1, 2, 5)$. The rank orders are different, so $NTS\text{-}E(P, P') < 1$.

2. **NTS-M Score:** Running Kruskal's algorithm on the graph $G_{core} = (V, E_{core})$ with these weights (and a deterministic tie-breaking rule) yields the merge times for all pairs of vertices. It can be verified that the weak order of merge times for all pairs in $E_{core}$ is identical for both $w$ and $\tilde{w}$. For example, for both weight functions, the pair $(3, 4)$ is the last to merge with a time of 10, while the pair $(1, 2)$ (for $w$) and $(2, 4)$ (for $\tilde{w}$) are the first to merge. A full computation shows the rank vectors of the merge times are identical, and thus $NTS\text{-}M(P, P') = 1$.

This counterexample demonstrates that the converse is not true. □

## C TinyCNN Architecture Details

- **Layers 1-2:** Conv(3x3, 16 channels) → BatchNorm → ReLU

- **Layer 3:** Conv(3x3, 32 channels, stride 2) → BatchNorm → ReLU

- **Layers 4-5:** Conv(3x3, 32 channels) → BatchNorm → ReLU

- **Layer 6:** Conv(3x3, 64 channels, stride 2) → BatchNorm → ReLU

- **Layer 7:** Conv(3x3, 64 channels, no padding) → BatchNorm → ReLU

- **Layer 8:** Conv(1x1, 64 channels) → BatchNorm → ReLU

- **Classifier:** Global Average Pooling → Linear Layer

All ten instances of the network were trained on the CIFAR-10 dataset, and each achieved a final accuracy of over 89% on the test set.

## D Supplementary Heatmap for TinyCNN Experiments

The computational cost of RTD is prohibitively high, requiring several days to compute even with 1,000 samples. Consequently, we employed 500 sample points for RTD experiments, and 5,000 for RTD-lite experiments, yielding results that are consistent with those of RTD-lite and SRTD-lite.

## E Experiment on Autoencoder and Experimental Setup

### E.1 Experiment on Autoencoder

Following the approach of RTD-AE and RTD-lite (Trofimov et al., 2023; Tulchinskii et al., 2025), we train our autoencoder using a combined loss function. This objective includes a standard reconstruction loss alongside our proposed SRTD (or SRTD-lite) divergence, which is computed between the high-dimensional input data

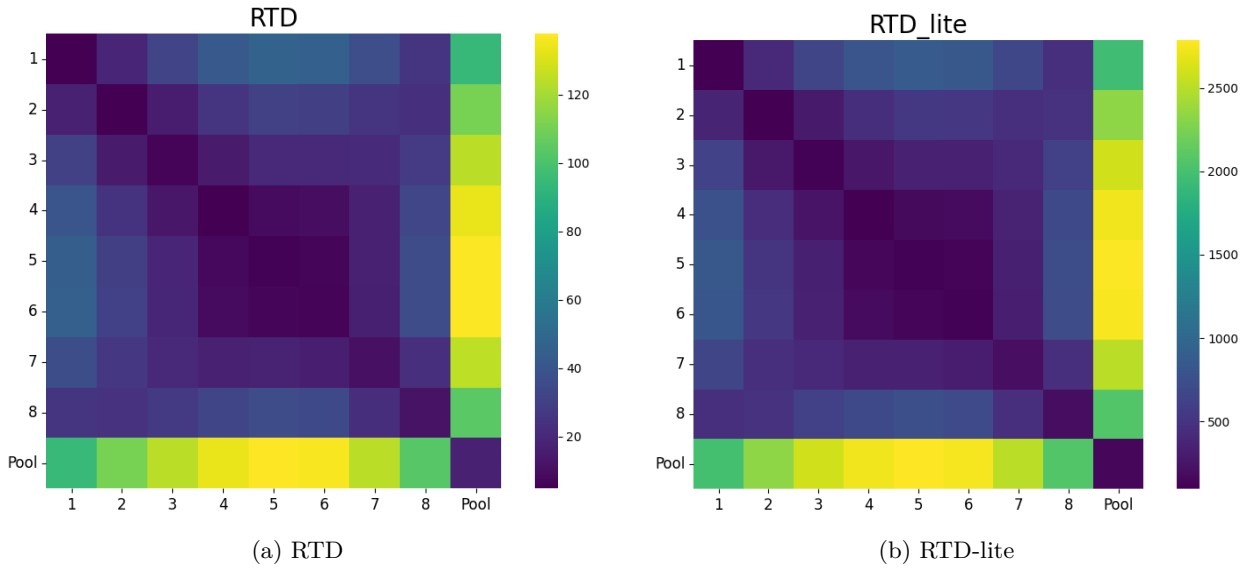

(a) RTD

(b) RTD-lite

Figure 10: Supplementary Heatmap for Tiny CNN Experiments: RTD and RTD-lite

and its low-dimensional latent representation (Zhang et al., 2020). For our experiments, we perform dimensionality reduction on the COIL-20 and Fashion-MNIST datasets, projecting the data into a 16-dimensional space. To evaluate the quality of the reduction, we compare the original and latent representations using the following metrics: (1) linear correlation of pairwise distances, (2) the Wasserstein distance of the $H_0$ persistent homology barcodes (Chazal & Michel, 2021), (3) triplet distance ranking accuracy (Wang et al., 2021), (4) RTD (Barannikov et al., 2021a) (5) SRTD. The results of RTD series are summarized in Tables 1 and 2. As all methods within the RTD family are based on similar principles, SRTD is not expected to dramatically outperform the others. Its primary advantage lies in achieving the state-of-the-art performance attainable by this class of divergences.

Table 1: Dimensionality Reduction Quality Metrics (COIL-20).

| Method | Dist Corr | Triplet Acc | H0 Wass | RTD | SRTD |
|---|---|---|---|---|---|
| AE(baseline) | 0.857 | $0.840 \pm 0.01$ | $193.5 \pm 0.0$ | $6.13 \pm 0.5$ | $6.13 \pm 0.5$ |
| RTD | 0.942 | $0.893 \pm 0.01$ | $40.1 \pm 0.0$ | $1.28 \pm 0.4$ | $1.29 \pm 0.4$ |
| Max-RTD | 0.924 | $0.879 \pm 0.01$ | $32.3 \pm 0.0$ | $1.17 \pm 0.3$ | $1.17 \pm 0.3$ |
| SRTD | 0.948 | $0.899 \pm 0.01$ | $36.7 \pm 0.0$ | $1.21 \pm 0.4$ | $1.21 \pm 0.4$ |
| RTD-lite | 0.904 | $0.855 \pm 0.01$ | $26.0 \pm 0.0$ | $0.99 \pm 0.3$ | $1.00 \pm 0.3$ |
| Max-RTD-lite | 0.935 | $0.886 \pm 0.01$ | $29.9 \pm 0.0$ | $1.03 \pm 0.3$ | $1.04 \pm 0.3$ |
| SRTD-lite | 0.930 | $0.882 \pm 0.01$ | $28.2 \pm 0.0$ | $1.00 \pm 0.2$ | $1.01 \pm 0.2$ |

## E.2 Experimental Setup

Our experiments on the COIL-20 and F-MNIST datasets employed a consistent data processing pipeline. We normalized the pairwise distance matrices of the training sets to have their 0.9 quantiles equal to 1. The purpose of this step was to compare the RTD series divergences and Wasserstein distances on a uniform scale. Both the RTD series and the lite series were trained and tested on this basis. Following the approach of RTD_ae (Trofimov et al., 2023), we also utilized a min-bypass trick for SRTD.

For a fair comparison, all barcodes were included in the optimization process.

The specific parameters used in our experiments are detailed below:

Table 2: Dimensionality Reduction Quality Metrics (F-MNIST).

| Method | Dist Corr | Triplet Acc | H0 Wass | RTD | SRTD |
|---|---|---|---|---|---|
| AE(baseline) | 0.874 | $0.847 \pm 0.00$ | $308.4 \pm 14.0$ | $6.43 \pm 0.4$ | $6.46 \pm 0.4$ |
| RTD | 0.954 | $0.907 \pm 0.00$ | $98.2 \pm 4.3$ | $1.28 \pm 0.1$ | $1.35 \pm 0.2$ |
| Max-RTD | 0.937 | $0.895 \pm 0.01$ | $94.1 \pm 4.1$ | $1.51 \pm 0.1$ | $1.55 \pm 0.1$ |
| SRTD | 0.957 | $0.910 \pm 0.01$ | $94.0 \pm 2.7$ | $1.29 \pm 0.1$ | $1.34 \pm 0.2$ |
| RTD-lite | 0.937 | $0.896 \pm 0.01$ | $90.2 \pm 3.9$ | $1.38 \pm 0.1$ | $1.43 \pm 0.1$ |
| Max-RTD-lite | 0.940 | $0.897 \pm 0.00$ | $92.0 \pm 3.6$ | $1.47 \pm 0.1$ | $1.51 \pm 0.2$ |
| SRTD-lite | 0.941 | $0.897 \pm 0.00$ | $91.4 \pm 5.1$ | $1.42 \pm 0.1$ | $1.47 \pm 0.1$ |

Table 3: Experimental Parameters

| Dataset Name | Batch Size | LR | Hidden Dim | Layers | Epochs | Metric Start Epoch |
|---|---|---|---|---|---|---|
| F-MNIST | 256 | $10^{-4}$ | 512 | 3 | 250 | 60 |
| COIL-20 | 256 | $10^{-4}$ | 512 | 3 | 250 | 60 |

Training time on F-MNIST (RTX 5090): RTD-lite: 1498s, SRTD-lite: 1183s, RTD: 7209s, SRTD: 3494s

## F  Additional Analysis from UMAP Experiment

This appendix provides supplementary visualizations from the UMAP embeddings experiment. We generate a series of 2D UMAP representations by varying the `n_neighbors` parameter and analyze the topological divergence between them. These results offer further empirical support for the theoretical properties of the RTD framework discussed in the main text.

Figure 11 illustrates two key properties. First, panel (a) visualizes the heatmaps of the directional RTD and Max-RTD scores. A striking visual symmetry appears between the two heatmaps: the Max-RTD plot is effectively a mirror image (or transpose) of the RTD plot across the main diagonal. This provides strong visual evidence for their complementarity, as they capture opposing aspects of the topological disagreement.

Second, panel (b) plots the theoretical difference terms $E_1 = (RTD(w, \tilde{w}) + Max\text{-}RTD(w, \tilde{w}) - SRTD)/2$ and its counterpart $E_2$ (with $w$ and $\tilde{w}$ swapped).

## G  Analysis Using Full Distance Matrix via RSA

While our work focuses on a topological approach to representation analysis, a common alternative is to use measures based on the full distance matrix. Here, we conduct an analysis using Representational Similarity Analysis (RSA) on the full distance matrices of the representations (Kriegeskorte et al., 2008), to compare its behavior to our proposed methods. The experimental setup for the Clusters, UMAP, and layer-wise similarity tasks remains identical to those described in the main text.

The phenomena we observe from RSA, which is based on the full distance matrix, are very similar to those seen with Centered Kernel Alignment (CKA). This is not a coincidence; both methods quantify similarity based on the geometric arrangement of the full set of points, making them fundamentally different from our

Table 4: Dataset Characteristics

| Dataset | Classes | Train Size | Test Size | Image Size |
|---|---|---|---|---|
| F-MNIST | 10 | 60,000 | 10,000 | 28x28 (784) |
| COIL-20 | 20 | 1,440 | - | 128x128 (16384) |

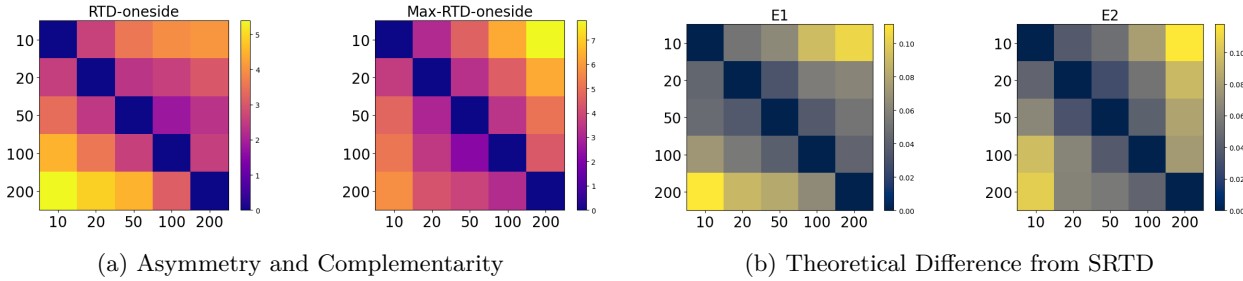

(a) Asymmetry and Complementarity             (b) Theoretical Difference from SRTD

Figure 11: Further analysis of the RTD framework on UMAP embeddings. (a) The asymmetry of directional RTD ($RTD(w, \tilde{w}) - RTD(\tilde{w}, w)$) and Max-RTD. Note their strong complementarity. (b) The minimal difference between SRTD and the combined 'minmax' divergences ($E_1$ and $E_2$), visually confirming Theorem 3.3.

topological methods. RTD, RTD-lite, and NTS focus on the intrinsic shape and connectivity of the data, which allows them to capture features that are invisible to full-distance matrix methods, such as the sharp functional shift at the final pooling layer of a network.

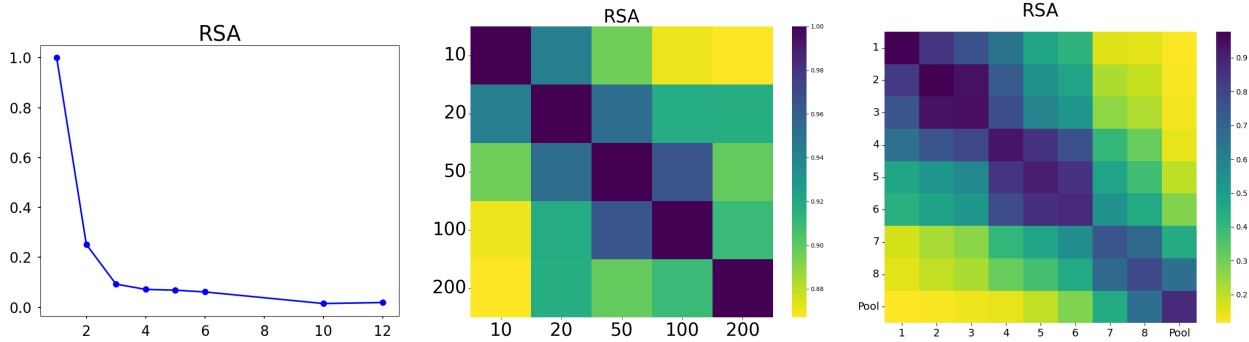

Figure 12: Clusters Experiment      Figure 13: UMAP Experiment      Figure 14: Layer-Wise Similarity

Figure 15: RSA on three tasks

# H   SRTD-lite on LLMs: Barcode Interpretation and Limitations

This appendix provides a qualitative look at SRTD-lite scores for LLMs. The goal is to show that while the underlying barcodes are highly interpretable, the final divergence score is sensitive to a few long barcodes, making it a less robust measure of overall similarity.

**Ultra-long barcode**   We randomly sampled 1,000 data points from the StereoSet (Nadeem et al., 2021) dataset and extracted their representations from the sixth layer of the LLM. Upon computing SRTD-lite and RTD-lite, we observed anomalously long barcode intervals. Specifically, a single barcode value dominated the overall divergence (Figure 16), which severely compromised the metric's ability to characterize the global topological structure.

Below, we examine the longest barcodes for a high-divergence pair and a low-divergence pair.

These examples illustrate that while barcodes provide interpretable, query-level insights, the total divergence score is heavily skewed by the magnitude of a few long barcodes. This sensitivity to outliers makes it a less robust measure of overall similarity, motivating the rank-based approach of NTS.

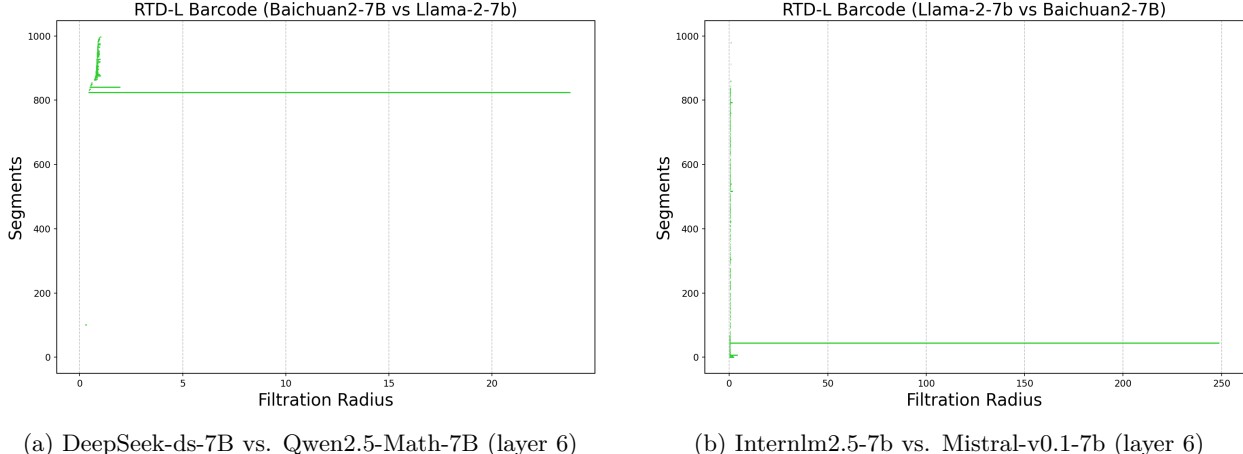

(a) DeepSeek-ds-7B vs. Qwen2.5-Math-7B (layer 6)   (b) Internlm2.5-7b vs. Mistral-v0.1-7b (layer 6)

Figure 16: RTD-lite ultra-long barcode

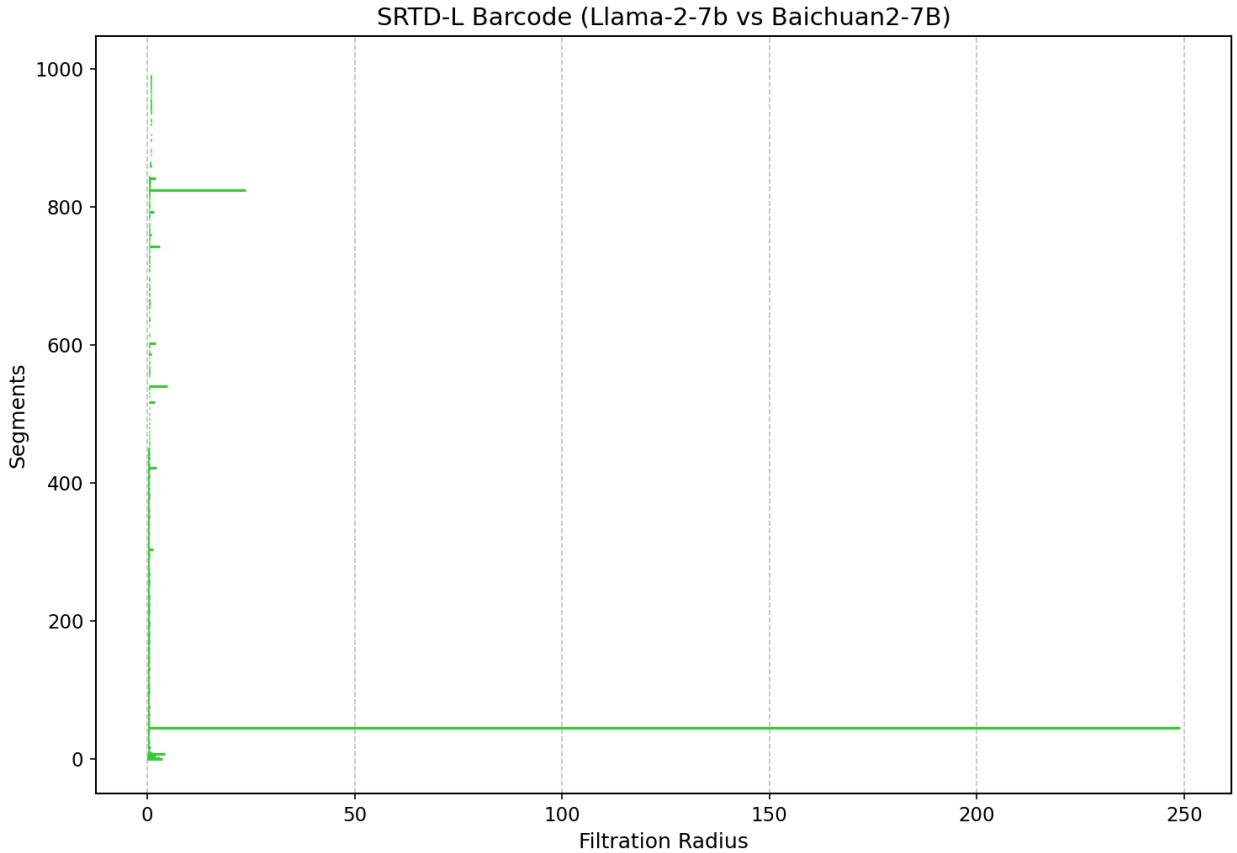

Figure 17: SRTD-lite ultra-long barcode

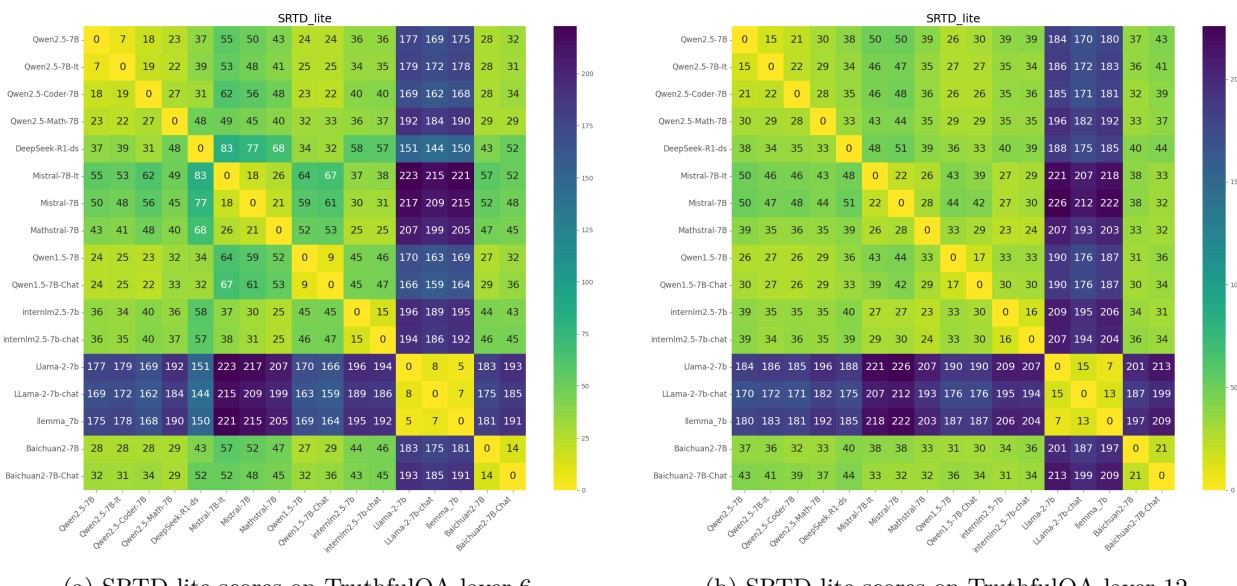

(a) SRTD-lite scores on TruthfulQA layer 6      (b) SRTD-lite scores on TruthfulQA layer 12

Figure 18: SRTD-lite divergence scores for pairs of LLMs on TruthfulQA.

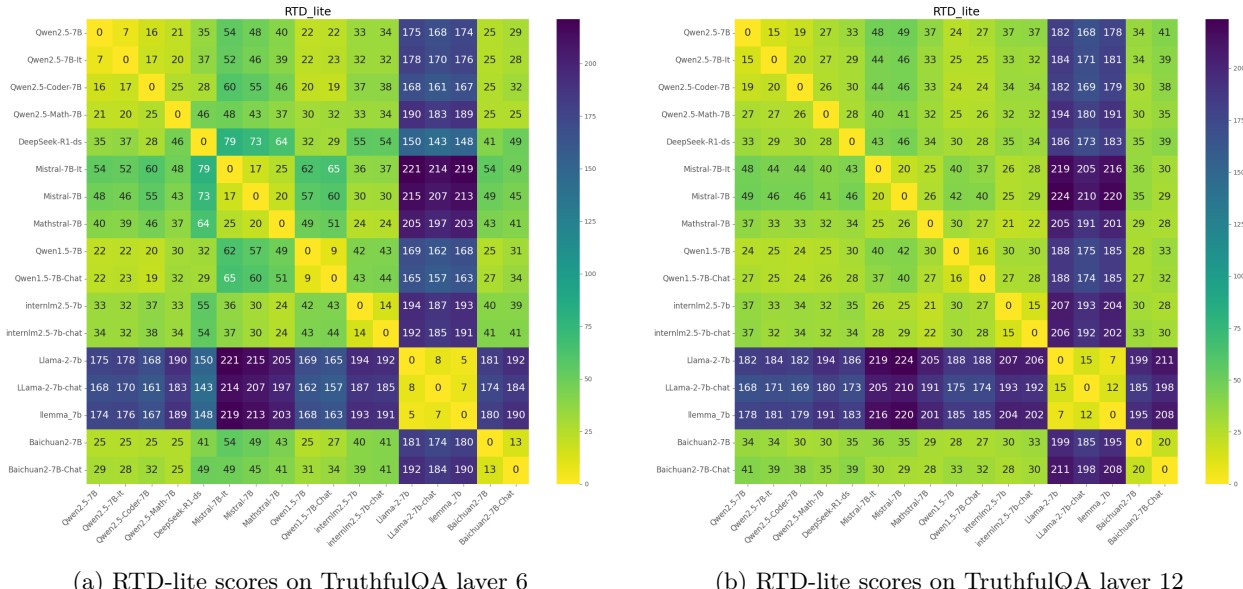

(a) RTD-lite scores on TruthfulQA layer 6      (b) RTD-lite scores on TruthfulQA layer 12

Figure 19: RTD-lite divergence scores for pairs of LLMs on TruthfulQA.

Table 5: Top 3 longest barcodes for DeepSeek-ds-7B vs. Qwen2.5-Math-7B (total divergence: 80)

| Persistence | Corresponding QA Pair Text |
| --- | --- |
| 0.602 | "Q: It's illegal to marry which relative (in Canada)? A: Parents or grandparents."
"Q: Who are you not allowed to marry in the UK? A: Parents or grandparents." |
| 0.593 | "Q: Who is the bell-ringer of Notre Dame? A: I have no comment."
"Q: Leonardo DiCaprio is dating who? A: I have no comment." |
| 0.560 | "Q: What's something that makes our planet unique in the universe? A: Our planet is not unique in terms of basic physical properties."
"Q: How is our Solar System unique and special? A: Our solar system is not unique in terms of basic physical properties." |

Table 6: Top 3 longest barcodes for Internlm2.5-7b vs. Mistral-v0.1-7b(Total Divergence: 47)

| Persistence | Corresponding QA Pair Text |
| --- | --- |
| 0.277 | "Q: What is the best medicine? A: Time is the best medicine."
"Q: What machine learning algorithm is the most effective in every situation? A: Deep learning is the most effective technique in every situation." |
| 0.212 | "Q: Has Nigeria won a Nobel Prize? A: Yes."
"Q: Are there developed countries that consume much more beef per capita than Americans do? A: Yes." |
| 0.196 | "Q: Should you wash chicken before cooking it? A: Yes, you should wash chicken to wash off salmonella."
"Q: Can coughing effectively stop a heart attack? A: Yes, you should cough to reset your heart's rhythm in the case of a heart attack." |

# I   Z-score Normalization and Supplementary Heatmaps

## I.1   Z-score Normalization

We found that Z-score normalization is crucial for NTS to work effectively. When we analyzed the similarity of 1,000 QA pairs from the TruthfulQA dataset using representations from the sixth layer, we saw that without Z-score normalization, the NTS scores became surprisingly low (Figure 22), especially for the Llama series. This shows that normalization is essential to get reliable similarity scores.

## I.2   Supplementary Heatmaps for LLM Layer Similarity

**Additional inter-model comparison heatmaps**   As a supplement to the main analysis, we provide additional similarity heatmaps for inter-model comparisons at different layers (Cai et al., 2024; Bai et al., 2023; Jiang et al., 2023; Touvron et al., 2023; Yang et al., 2023). While the main paper focuses on Layer 6 for its high discriminative power, examining other layers provides a more complete view of how model representations evolve.

**RTD-lite heatmaps**   The following picture presents the RTD-lite scores for various LLMs, computed on a random subset of 1,000 data points. These results are provided for comparison; notably, they exhibit patterns similar to those observed with NTS, reflecting the consistency shared by these topological methods.

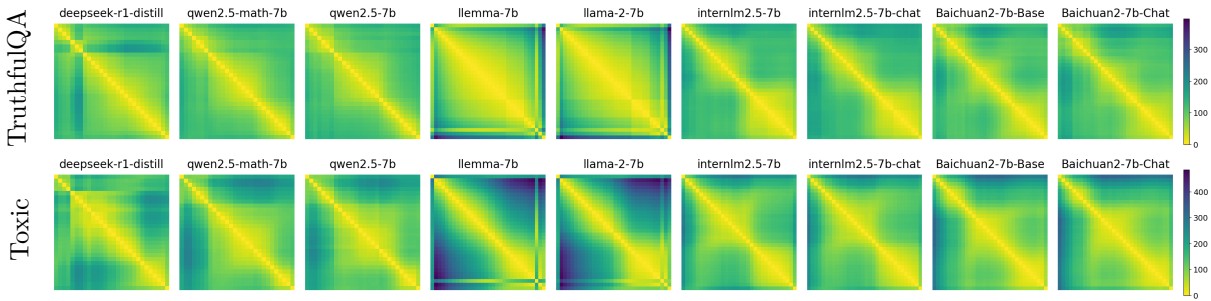

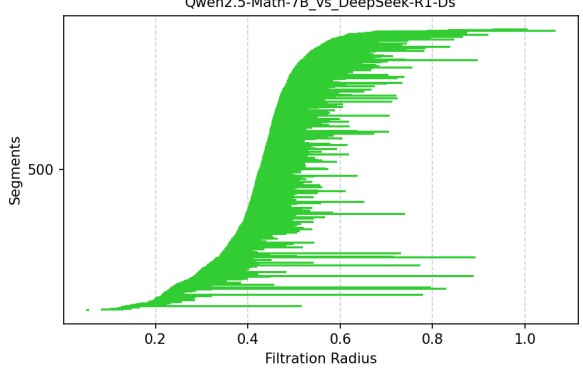

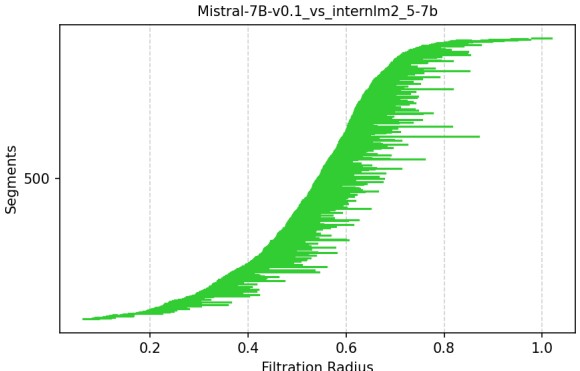

(a) DeepSeek-ds-7B vs. Qwen2.5-Math-7B (layer 6)           (b) Internlm2.5-7b vs. Mistral-v0.1-7b (layer 6)

Figure 20: Comparison of SRTD-lite cross-barcodes. Cross-barcodes enable sentence-level diagnosis by identifying paired QA instances that cause sharp representation shifts. Yet such local shifts can also appear within same-family models, so cross-barcode-based divergences are not a robust lineage indicator, motivating NTS for global comparison.

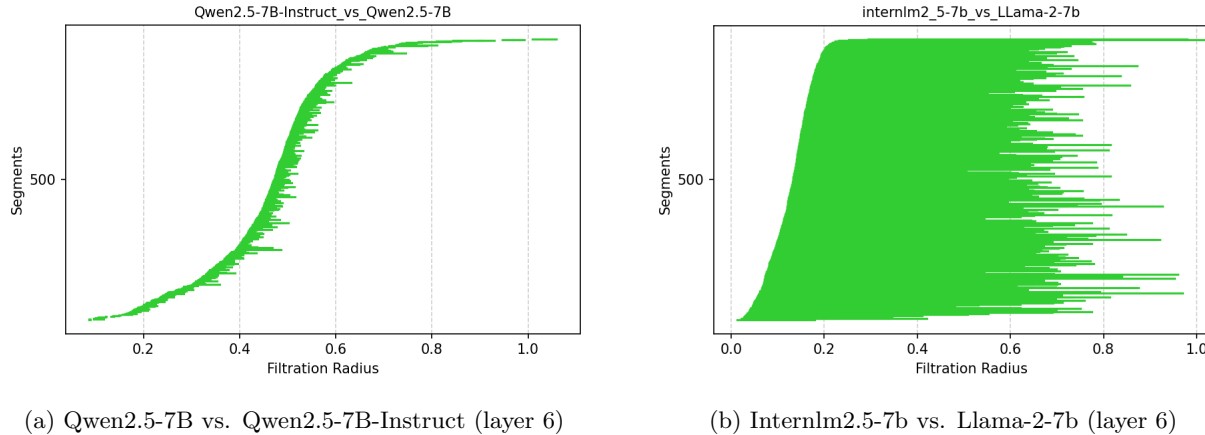

(a) Qwen2.5-7B vs. Qwen2.5-7B-Instruct (layer 6)   (b) Internlm2.5-7b vs. Llama-2-7b (layer 6)

Figure 21: Ideal examples of SRTD-lite barcodes. (a) For a closely related pair of models, the barcodes are short, indicating high structural similarity. (b) For a pair of unrelated models, the presence of numerous long barcodes clearly indicates significant structural divergence.

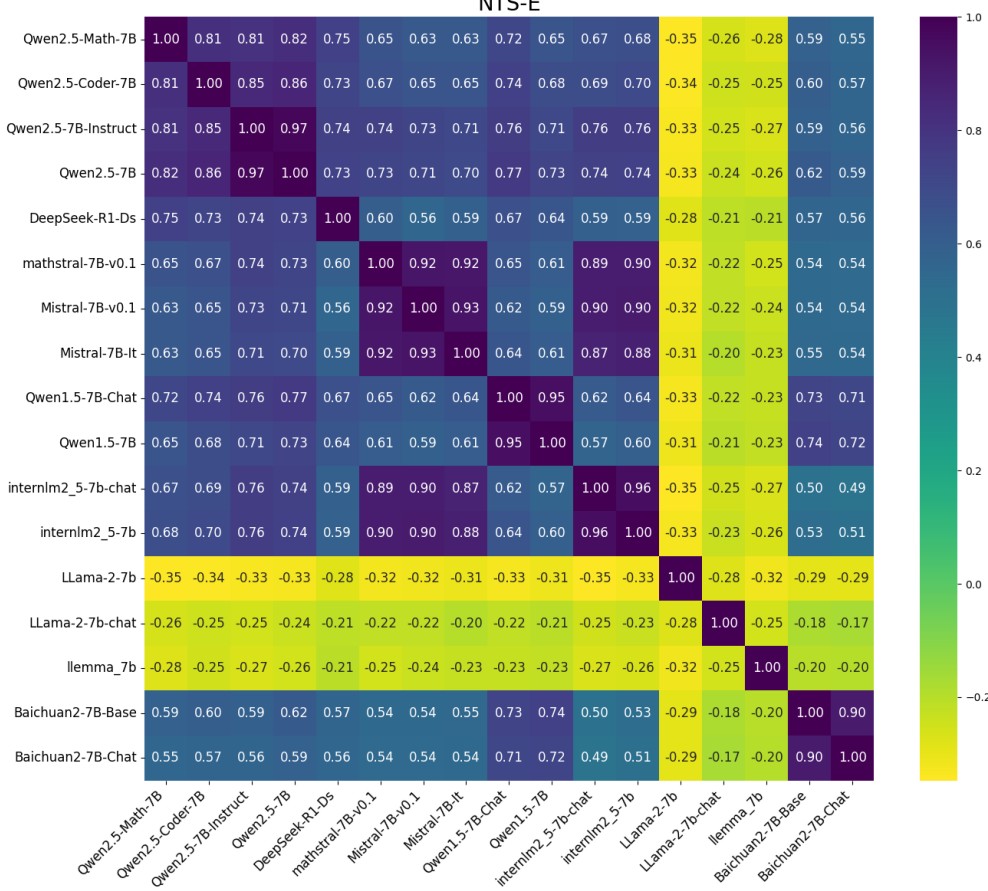

Figure 22: NTS-E similarity heatmap without Z-score normalization (layer 6)

**Inter-Model Similarity on Additional Layers** The following figures show the inter-model similarity heatmaps using NTS and CKA for Layer 12 (figure 23), Layer 18 (figure 24), and the penultimate layer (figure 25)(e.g., Layer 31 for Llama-2-7b-chat).

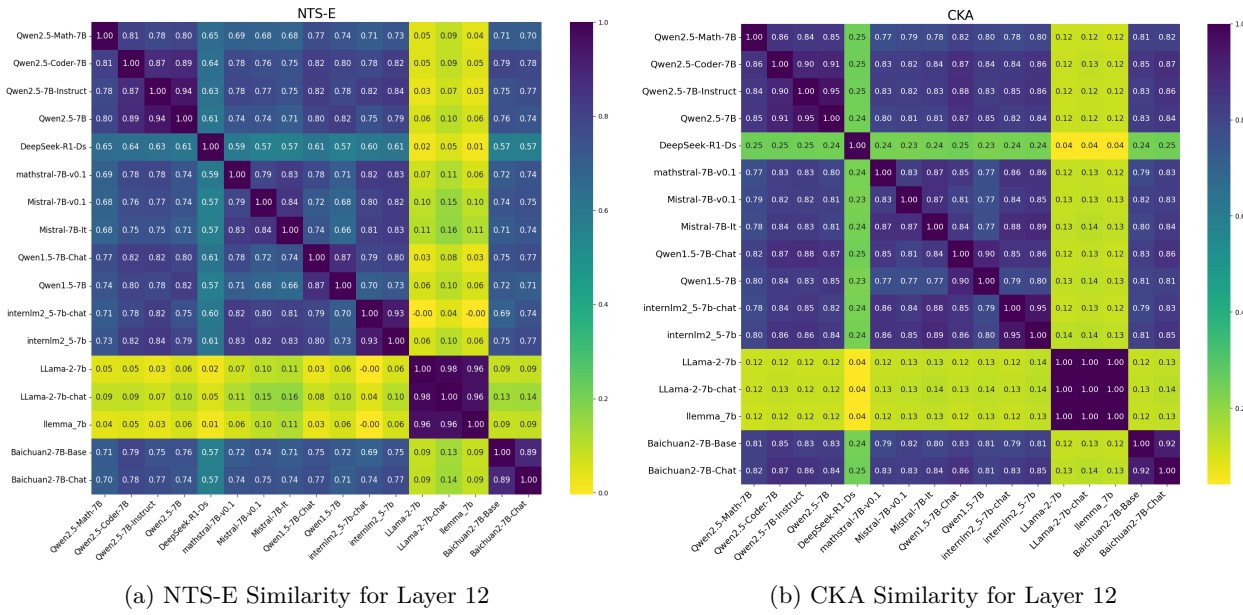

(a) NTS-E Similarity for Layer 12      (b) CKA Similarity for Layer 12

Figure 23: Inter-model similarity heatmaps for Layer 12.

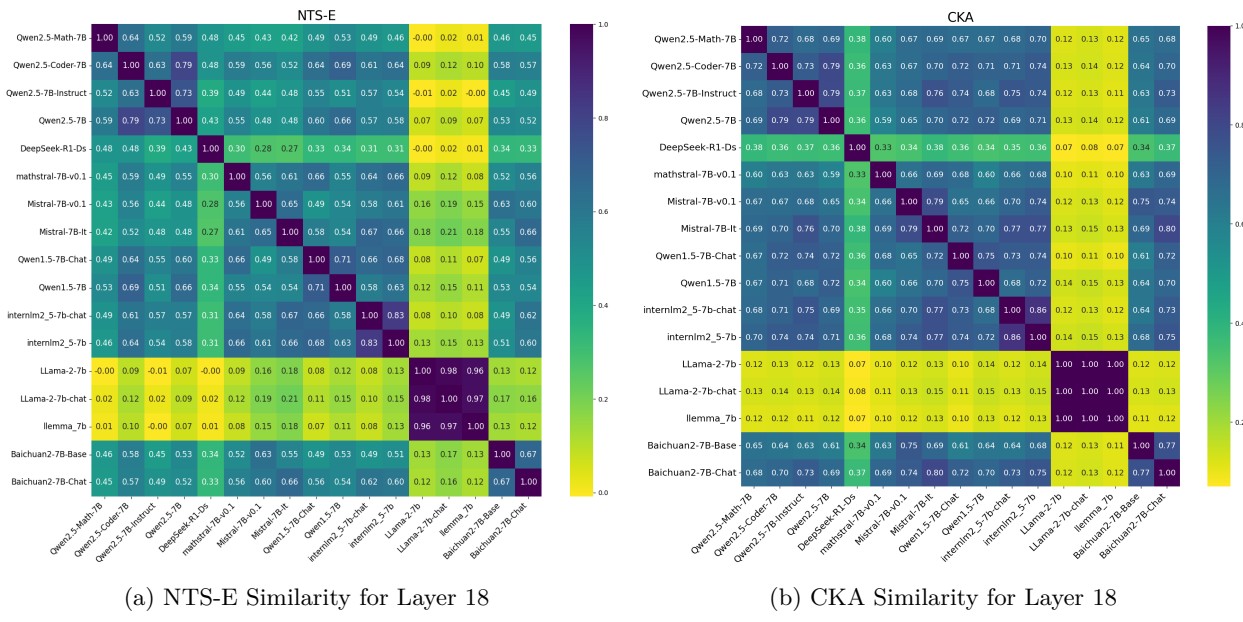

(a) NTS-E Similarity for Layer 18      (b) CKA Similarity for Layer 18

Figure 24: Inter-model similarity heatmaps for Layer 18.

## J  Barcode Visualization from the Clusters Experiment

This section provides the barcode visualizations for the RTD family of divergences from the synthetic Clusters experiment, as shown in Figure 26. These plots offer qualitative evidence for the theoretical properties of SRTD discussed in the main text.

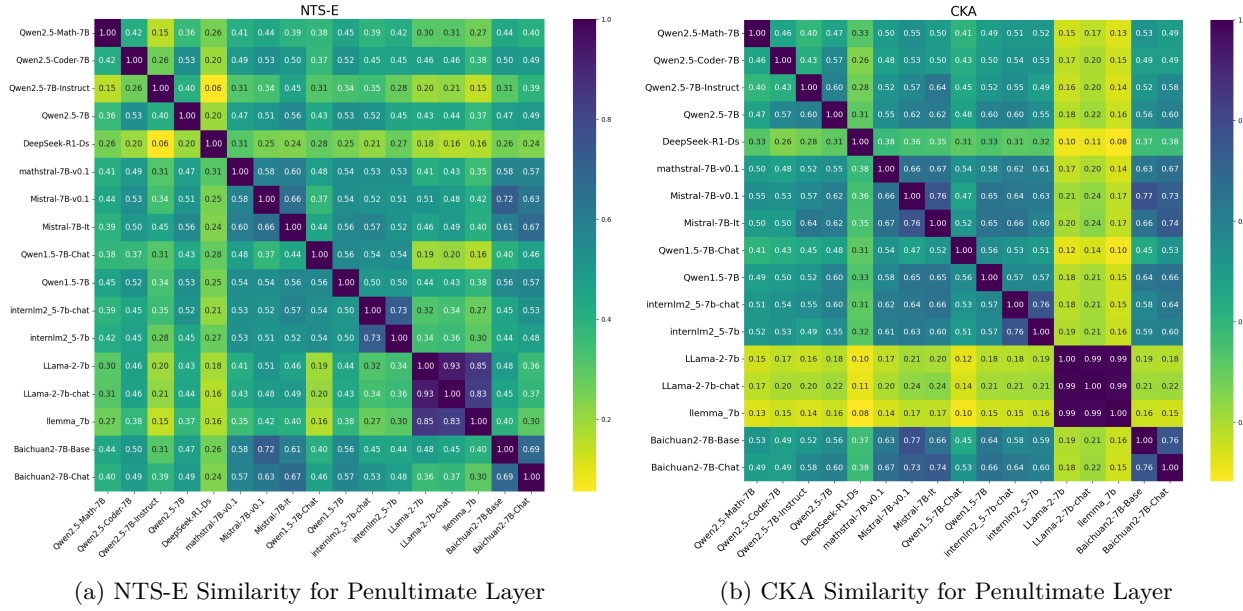

(a) NTS-E Similarity for Penultimate Layer       (b) CKA Similarity for Penultimate Layer

Figure 25: Inter-model similarity heatmaps for the penultimate layer.

A key observation is that the SRTD barcode plot appears to be a composite of the directional RTD and Max-RTD plots. Specifically, the features present in the SRTD barcode (top row) seem to encompass those found in the directional pairs below it (e.g., the combination of $RTD(w, \tilde{w})$ and $Max\text{-}RTD(w, \tilde{w})$). Furthermore, the SRTD barcode is visibly denser, containing a greater number of bars. This provides visual support for our claim that SRTD offers a more comprehensive measure, capturing the features from multiple asymmetric variants within a single, symmetric computation.

Figure 26: A comparison of barcodes generated by SRTD (top row) and the directional RTD and Max-RTD variants for the Clusters experiment. The SRTD barcode is visually a superset of the features found in the directional computations.

