# OpenReview forum: "Symmetric Divergence and Normalized Similarity: A Unified Topological Framework for Representation Analysis"
_TMLR — Accepted by TMLR_

### Review · Reviewer_uPvM · 2026-03-06

**Summary Of Contributions:**

The authors propose a unified topological framework for comparing neural network representations to address the limitations of existing unnormalized and asymmetric topological divergences. Specifically, it introduces Symmetric Representation Topology Divergence (SRTD) to resolve the theoretical asymmetry of prior RTD methods, and Normalized Topological Similarity (NTS), a rank-based, scale-invariant metric bounded between -1 and 1 for robust benchmarking.

**Key strengths**

I think the paper is interesting and addresses a meaningful problem. The proposed toolkit appears computationally efficient and shows strong empirical utility. In the experiments, it effectively captures functional architectural shifts in CNNs and maps the genealogies of LLMs, where standard geometric baselines such as CKA tend to fail due to distance saturation.

**Key weakness**

One limitation, in my view, is that the proposed methods rely solely on 0-dimensional connected components. As a result, they sacrifice higher-dimensional topological features (e.g., $H_1$ and $H_2$), which may limit the ability to capture richer structural information in the representations.

**Audience:**

Yes

**Audience Explanation:**

In my opinion, researchers working on representation learning, interpretability, and foundation model evaluation will likely find the proposed NTS and SRTD metrics useful. The paper addresses a known limitation of standard geometric approaches such as CKA, namely the “distance saturation” issue when analyzing deep neural network representations. In addition, the proposed scalable and mathematically grounded topological framework appears particularly useful for analyzing the lineage and functional shifts of large language models, which could provide practical value for the broader ML community.

**Broader Impact Concerns:**

I have not found any discussions about the limitations and potential negative societal impact. But in my opinion, this may not be a problem, since the work only focuses on the topological analysis of neural networks. Still, it is highly encouraged to add corresponding discussions.

**Claims And Evidence:**

Yes

**Claims Explanation:**

**Theoretical Claims**

I find the theoretical demonstration to be well presented. The authors provide rigorous mathematical proofs (e.g., Theorem 3.3 and Corollaries 3.4/3.5) to formally establish the relationship between directional RTD, Max-RTD, and the proposed SRTD. In my view, this theoretical analysis can supports their claim that SRTD resolves the heuristic asymmetry present in prior RTD-based methods.

**Empirical Validation**

I think the experimental design is convincing. The evaluation progresses from controlled synthetic setups (e.g., Clusters and UMAP) to practical large-scale deep learning scenarios. For instance, the LLM experiments clearly demonstrate that NTS can mitigate the “distance saturation” issue that often affects CKA. The evidence is particularly interesting when NTS successfully identifies the genealogical relationship between the distilled DeepSeek-R1-Distill model and its Qwen parent, a connection that the baseline CKA fails to detect.

**Efficiency Claims**

The claims regarding computational scalability are also supported by empirical runtime benchmarks. From the reported results, NTS-E and SRTD-lite scale significantly better than the baseline RTD-lite as the sample size increases.

**Requested Changes:**

1. Can you give some discussion with the paper “Quantitative Performance Assessment of CNN Units via Topological Entropy Calculation”?
It would be helpful. since this paper also focuses on charaterizing the the geometric representation of networks via topological analysis.

2. It should be interesting to use this comparison to explicitly discuss the trade-offs of your approach. While the MST-based framework appears to work very well for scalable, model-to-model comparisons, relying only on 0-dimensional features inevitably sacrifices some fine-grained information. For example, it may not capture the unit-level interpretability and higher-order topological structures (e.g., $H_1$, $H_2$) that methods such as topological entropy are designed to analyze.

Some typos:
Figure 9. "Runing" -> "Running"

---

> ### Author Response · Authors · 2026-03-18
>
> We deeply appreciate your highly positive evaluation, particularly your recognition of our theoretical proofs, empirical validation on LLMs, and computational scalability.
>
> **Key Weakness & Requested Change: Limitation of 0-dimensional features and trade-off discussion.**
> We fully agree with your insightful observation. Relying primarily on 0-dimensional features is a fundamental trade-off: it achieves the high scalability needed for massive macro-level LLM benchmarking, but inevitably sacrifices fine-grained, higher-order topological insights and unit-level interpretability. Following your excellent suggestion, we have incorporated "Quantitative Performance Assessment of CNN Units via Topological Entropy Calculation" (Zhao et al., 2022) into our revised manuscript. In both the Introduction and Conclusion (Section 7), we now discuss this trade-off more explicitly in relation to prior topological approaches: while topological entropy is well suited to microscopic, unit-level evaluation, our macroscopic approaches instead emphasize highly efficient global comparisons.We have highlighted these additions in blue in Section 1 (Introduction) and Section 7 of the revised manuscript.
>
> **Requested Change: Typo in Figure 9.**
> Thank you for catching this. We have corrected "Runing" to "Running" in Figure 9.
>
> **Requested Change: Broader Impact Concerns.**
> We have added a dedicated Broader Impact Statement section to the paper. We clarify that while our work focuses on the theoretical analysis of representations and carries no direct negative societal impact, it indirectly supports AI safety by providing tools to enhance the transparency and structural evaluation of large foundation models.This newly added section has been highlighted in blue at the end of the revised manuscript.

---

### Review · Reviewer_SHgq · 2026-03-12

**Summary Of Contributions:**

**Contributions**
1. The paper introduces a modification of RTD, called SymRTD (SRTD). In opposite to RTD, it is symmetric by definition and doesn't required extra symmetrization. The same extension is done for RTD-Lite.
2. Theoretical relationship between RTD, Max-RTD, and SRTD is established.
3. Experimental study shows benefits of SRTD, NTS over other representation comparison measures is some practical cases.

**Strengths**
1. An introduction for a reader not familiar with TDA is provided.
2. The language of the paper is fine, main ideas are easy to follow. Algorithmic details are explained.
3. Paper includes both theory (Theorem 3.3, Theorem 4.1) and experimental results.

**Weaknesses**
1. The weakness of NTS is its non-differentiability.
2. In Fig. 5, 6  the superiority of SRTD w.r.t. of RTD is not proved. Also, the illustration of RTD in the "UMAP Embeddings Experiment" is missing.
3. Section 5.3: As I understand, author claim that the monotony of NTS is its main advantage. But why? It is not clear a priori that the dissimilarity of representation in layers must change monotone. Please consider adding here more arguments, maybe visualization, etc.
4. In Section 5.4, a comparison with RTD and RTD-Lite is not provided.

**Audience:**

Yes

**Audience Explanation:**

Study of DNN internals is of high importance since still many aspects of their functioning are not fully understood.

**Claims And Evidence:**

Yes

**Claims Explanation:**

Yes, experimental results are sound and claims made in submission are supported.

**Requested Changes:**

Please address "questions" above. However, I think that these issues are mostly about clarifications and don't undermine main contributions of the paper.

---

> ### Author Response · Authors · 2026-03-18
>
> We sincerely thank you for the constructive feedback and for recognizing our theoretical contributions and experimental soundness. Below is our point-by-point response:
>
> **Weakness 1: Non-differentiability of NTS.**
> We completely agree. In Section 7, we explicitly identify non-differentiability as the primary limitation of NTS. Our unified framework is designed to serve complementary needs: SRTD preserves differentiability and serves as an effective loss for model optimization, while NTS provides a bounded, scale-invariant metric necessary for robust cross-scenario benchmarking.
>
> **Weakness 2 & 3: Objectives of Fig 5 & 6, missing UMAP RTD illustration, and monotonicity.**
> We would like to clarify the experimental objectives of Figures 5 and 6, which aim to highlight the advantages of our proposed framework over geometric baselines (CKA) and unnormalized divergences, rather than to prove SRTD's superiority over RTD:
>
> - **Regarding Fig 5 (UMAP) and missing RTD:**
>   The goal is to show that our framework captures structural changes where CKA fails. We used SRTD-lite as the representative measure. Instead of redundant plots, we provided the directional RTD and Max-RTD heatmaps in Appendix F (Figure 11). Figure 11(a) visually proves their strong asymmetry and inherent complementarity, perfectly justifying the theoretical motivation to unify them into SRTD.
>
> - **Regarding Fig 6 (TinyCNN) and the "monotonicity" expectation:**
>   The expectation of smooth, monotonic similarity decay across layers is a well-established empirical property of hierarchical feature learning in CNNs. Figure 6 demonstrates that NTS is superior because it successfully recovers this monotonic hierarchy (which unnormalized SRTD-lite fails to do), while also detecting the sharp functional shift at the pooling layer—a crucial topological feature completely missed by CKA.
>
> **Weakness 4: Missing comparison with RTD/RTD-Lite in Section 5.4 (LLMs).**
> Regarding the missing comparison with the full RTD: Computing the full RTD for the $N=1000$ samples used in our LLM evaluation is computationally prohibitive, requiring several days of computation (as noted in Appendix D). However, because RTD-lite was specifically designed for scalability and consistently yields results highly correlated with the full RTD, it serves as a robust and reliable proxy for this evaluation. We have provided the full LLM inter-model similarity heatmaps for RTD-lite in Appendix H (Figures 18 and 19). As analyzed there, unnormalized divergences like RTD-lite struggle in LLM settings because the total score is heavily skewed by a few anomalous "ultra-long barcodes". This extreme sensitivity to outlier pairs makes them unreliable for mapping global model relationships, which directly motivated our design of the rank-based NTS.

---

### Review · Reviewer_dpNg · 2026-04-18

**Summary Of Contributions:**

The paper proposes SRTD as a symmetric topological data analysis metric between two point clouds $P \in \mathbb{R^{N \times D_1}}$ and $P' \in \mathbb{R^{N \times D_2}}$.
RTD measures topological structural (specifically, connected components in $0^{th}$ dimension) differences between $P$ and $P'$ by using auxiliary matrix $M_{min}$ and is asymmetric by definition.
The proposed SRTD addresses this asymmetric nature of RTD by using $M_{sym}$ formed by replacing $w$ by $min(w, w')$ in the Max-RTD auxiliary matrix. This is equivalent to measuring the difference between the topological structures present between $\mathcal{G}^{max(w, w')}$ and $\mathcal{G}^{min(w, w')}$, computed at all thresholds $\alpha$.
The authors also propose NTS, a bounded measure of topological similarity between $P$ and $P'$, computed as a correlation between the edge vectors of MST(P) and MST(P') that can be used for benchmarking cross-scenarios.

### Strengths
1. SRTD addresses the assymteric nature of RTD and computes R-cross-barcode once instead of twice as required for RTD or Max-RTD. Smilarly SRTD-lite is more efficent than RTDL and Max-RTDL.
2. The authors have proposed NTS as a bounded measure of topological similarity between point clouds. NTS computes the correlation between the edge weight vectors of $P$ and $P'$ over the union of their MST, resulting in a value between $[-1, 1]$, that could be used for comparing cross-scenarios.
3. The authors have evaluated SRTD and NTS in diverse scenarios - synthetic cluster dataset, small CNN models trained on F-MNIST, LLMs on TruthfulQA, and the ToxicGen dataset.

### Weaknesses
1. STRD is computationally efficient but does not seem to be better than other RTD variants.
2. The efficacy of NTS as a normalized score for comparing different scenarios is not experimentally validated enough. Please see 'Requested Changes' section.

**Additional Comments:**

The paper proposes a set of novel metrics - SRTD and NTS- that are designed to efficiently quantify underlying topological differences between neural representations and are valuable tools, particularly in the study of latent spaces and model alignment.
The theoretical motivations for SRTD and NTS are well-articulated, and the move toward bounded, symmetric measures is a significant practical improvement. While I am inclined toward recommending acceptance, I believe a simple experiment is needed to corroborate the efficacy of NTS as a normalized cross-scenario metric, and clarifications/discussions should be provided for certain experimental findings as mentioned above.

**Audience:**

Yes

**Audience Explanation:**

The paper introduces SRTD (Symmetric Representation Topological Distance) and NTS (Network Topological Similarity), which aim to address the computational and structural limitations of existing topological comparison methods. SRTD resolves the inherent asymmetry and requires only a single computation of the R-cross-barcode—as opposed to the dual computations necessitated by RTD or Max-RTD. The proposal of NTS as a bounded measure ([−1,1]) is a significant practical contribution for comparing different scenarios. Thus, collectively these metrics provide a robust framework for investigating the structural alignment of latent spaces, model representations, and datasets. Given the increasing focus on model interpretability and representation alignment in machine learning, these tools are likely to be of significant interest to the community.

**Claims And Evidence:**

Yes

**Claims Explanation:**

The authors have performed the comparison of SRTD with RTD versions by evaluating and comparing the divergence score on a synthetic cluster dataset. The computational time of SRTD is compared with RTD versions on CIFAR-10. Furthermore, the efficacy of SRTD in training AE is compared to other RTD-variants. The normalized score given by NTS is used to compare the representations of different layers of TinyCNN and the Transformer model, as well as to compare different LLMs. The NTS findings are then compared with CKA.

While the scope of the evaluation is thorough, there are specific experimental findings that require further explanation to fully support the authors' conclusions (see 'Requested Changes' section)."

**Requested Changes:**

I would like to suggest the following changes, and need further clarification on the posed questions.

1. **Empirical Validation of NTS as a Cross-Scenario Metric**: As of now, the efficacy of NTS as a normalized score for comparing different scenarios is not experimentally validated. For example, there should be a controlled experiment where multiple scenarios are created by changing the underlying structure systematically and then evaluating both NTS and RTD for each scenario. After that, doing some kind of correlation study between the normalized score given by NTS and RTD score can corroborate that NTS can be used to compare/rank cross-scenarios. The existing experiments don't validate it because -
     1. In the synthetic cluster experiment, the decreasing NTS and increasing RTD plots show different rates of change, which complicates interpretation. So a correlation study on NTS and RTD score is needed.
    2. The existing experiments on measuring similarity between layer representations or models lack a well-defined ground truth pattern that NTS is expected to validate.

3. **Sensitivity and Skewness of NTS (Figure 4c)**
     The sensitivity of NTS appears highly non-linear, which may undermine its use as a normalized score.
      In Figure 4c, NTS drops sharply when the number of clusters changes from 1 to 2, but for a larger number of clusters, the change in NTS value is very slow. Specifically, the drop in NTS value is huge (from 0.8 to 0.3) when #clusters changes from 1 to 2, but the change in NTS value is relatively small when #clusters changes from 2 to 12. If NTS is intended to be a reliable comparative measure, how do you account for this "saturation" or skewness? Is a change in NTS value meant to be proportionate to the structural change?

      As another simple example, consider an alternate scenario where Rep A is exactly as shown in Figure 3, but Rep. B in Figure 3 is changed. Let the change be such that node 'a' is labeled as node 'b', 'b' is labeled as 'a', and so on, i.e., renaming points as a->b, b->c, c->d, d->e, and e->a. Here, we don't change any distance between nodes other than renaming them so the order of connections in Rep B remains the same: 1. (b,c), 2. (e,a), 3. (c,d), and 4. (d,e). In this scenario, in Rep A, node 'a' is closer to 'b' than node 'e', while in Rep B, node 'a' is closer to 'e' than 'b'. However, in $H_0$, the number of connected components still remains unchanged for the same threshold $\alpha$, but the NTS score is likely to change owing to a change in edge vectors used for computing correlation. So, in this scenario, is any change in NTS value desired, and if so, does it reflect a topological change or geometric change?

4. **NTS-E vs. NTS-M**

   Please clarify the notation in Equation 2, specifically confirming if $T_w$​ refers to the unique path in the MST.

   NTS-E is based on distance $w_{ij}$, while NTS-M applies a local max operation over paths connecting $i$ and $j$. So how do NTS-E and NTS-M relate? Can you discuss the boundary cases where NTS-E and NTS-M are equivalent, versus cases where one acts as a stricter constraint than the other?

5. **Interpretability of the "Functional Shift" (Figure 6)**
> Only the topological measures (b–d) capture the sharp structural break at the final pooling layer—a functional shift missed by geometric analysis.

   In Figure 6, the NTS and SRTD plots show huge divergence between the pool layer and any other layer of the model. In fact, NTS (SRTD) between the pool layer and layer 8 is very small (large) relative to any other pair of layers. This is counterintuitive, as the input to the pooling layer is the output of layer 8 itself, and the output of the pool layer is the locally averaged output of layer 8. Then how can the underlying topological structures be so different? In fact, one would expect high similarity as captured by CKA. Why do the proposed topological metrics fail to capture the similarity between Layer 8 and its pooled output? What is the specific "functional shift" that the authors claim is being captured here that CKA is missing?

   Minor fix: Please synchronize the colorbar scales for plots (b) and (c) to [−1,1] to allow for an objective visual comparison of NTS-E and NTS-M.

2. > When $RTD(w, \tilde{w}) > RTD(\tilde{w}, w)$, we consistently find that $Max-RT D(w, \tilde{w}) < Max-RTD(\tilde{w},w)$. This suggests that the topological structural differences between $R_{\alpha}(\mathcal{G}^w) \cup R_{\alpha}(\mathcal{G}^{\tilde{w}})$ and $R_{\alpha}(\mathcal{G}^w) \cap R_{\alpha}(\mathcal{G}^{\tilde{w}})$ seem to be the core reason for the asymmetry in RTD.

    In Table 4f, the asymmetric values of RTD and Max-RTD seem to be negatively correlated. That is, the decreasing and increasing trend in the asymmetric values of the RTD and Max-RTD with an increase in cluster numbers seems to be similar. This negative correlation seems to be because of the design of the auxiliary matrix. So, how does this support the above-quoted claim?

2. > The results confirm that SRTD and SRTD-lite achieve top-tier performance on quality metrics while being faster than their predecessors.

    Use of SRTD during optimization is not clearly supported. In Table 1, SRTD performs worse than Max-RTD, and SRTD-lite slightly lags behind RTD-lite on metrics - RTD and SRTD. However, one would expect that AE trained on SRTD-based loss function should result in the best performance on at least the SRTD metric during evaluation. What could be the explanation for this discrepancy? Also, please provide the performance plot of intermediate checkpoints during training with different loss functions on RTD and SRTD metrics?

3. In Figure 4, SRTD results in a similar increasing trend as Max-RTD. Similarly, SRD-lite shows a similar increasing trend as Max-RTD-lite. Since these scores are unbounded, one can compare them only relatively, and as observed from the plot, SRTD offers similar findings to Max-RTD. Similar is the case between RTD and SRTD. So, is it correct to say that SRTD results in similar output as asymmetric Max-RTD/RTD and is only better in terms of computational time?

6. In Figure 7, most of the plots generated by NTS and CKA show similar patterns except for models - DeepSeek, llemma, llama, and baichaun. Similarly, in Figure 8, the CKA score of the DeepSeek model with all other models is very small, while NTS marks it relatively similar to all other models. Have you conducted additional analysis to explain this discrepancy? Does DeepSeek possess a unique geometric representation (captured by CKA) that nonetheless shares the same underlying topological backbone (captured by NTS)?

---

> ### Author Response · Authors · 2026-05-01
>
> We sincerely thank the reviewer for the constructive feedback. Below is our point-by-point response.
>
> ## Q1: Empirical validation of NTS as a cross-scenario similarity score
>
> We thank the reviewer for this important suggestion. The reviewer asks whether NTS is suitable for cross-scenario comparison, and whether there is a clear case where NTS gives the expected judgment while RTD-style divergences give a counterintuitive one. To address this directly, we added a two-pair stress experiment with an explicit expected ordering.
>
> Both pairs have 80 samples in 50 dimensions. The first pair, $A,B$, is structurally related: $A$ is a clustered point cloud with 8 clusters and 10 samples per cluster, and $B$ is obtained from $A$ by cluster-wise anisotropic warping. Sample identity and cluster-level genealogy are preserved, although the geometry is strongly distorted. The second pair, $C,D$, consists of two independently generated high-dimensional shell point clouds with random sample-wise pairing. Thus, $C,D$ has a formal correspondence, but this correspondence does not reflect shared structure. PCA and correspondence visualizations are provided in the supplementary material for intuition; all quantities are evaluated in the original 50-dimensional space.
>
> A reasonable cross-scenario similarity score should rank the related pair $A,B$ as more similar than the randomly paired unrelated pair $C,D$. Over 20 random seeds, we obtain:
>
> | Quantity | Related $A,B$ | Randomly paired $C,D$ | Correct ranking |
> |---|---:|---:|---:|
> | NTS-E $\uparrow$ | $0.2899\pm0.0825$ | $-0.7148\pm0.0299$ | 20/20 |
> | NTS-M $\uparrow$ | $0.4741\pm0.0907$ | $-0.2285\pm0.0908$ | 20/20 |
> | RTD $\downarrow$ | $8.8176\pm0.3560$ | $4.8341\pm0.2123$ | 0/20 |
> | SRTD $\downarrow$ | $17.6656\pm0.7124$ | $12.6268\pm0.4543$ | 0/20 |
> | RTD-lite $\downarrow$ | $8.6933\pm0.3588$ | $1.5122\pm0.1247$ | 0/20 |
> | SRTD-lite $\downarrow$ | $17.4232\pm0.7214$ | $7.3589\pm0.3339$ | 0/20 |
>
> This is the key cross-scenario result. NTS-E and NTS-M correctly rank the related warped pair $A,B$ as more similar than the unrelated random pair $C,D$ in all seeds. In contrast, RTD, SRTD, RTD-lite, and SRTD-lite all assign smaller divergence to $C,D$ than to $A,B$, reversing the expected ordering.
>
> This reversal occurs because RTD-family quantities are unbounded absolute divergences. They compare barcode lengths or MST-weight differences after normalizing each distance matrix by its own $0.9$ quantile. This fixes one reference percentile, but it does not make heterogeneous edge-length distributions fully comparable. In the shell pair $C,D$, pairwise distances are homogeneous and concentrated, so after quantile normalization the aggregate MST/barcode differences can be small. In the clustered pair $A,B$, the distance distribution has a much stronger long-short contrast: intra-cluster edges are short, inter-cluster edges are long, and the anisotropic warping changes both local and global connections. Thus, RTD/SRTD can assign a larger absolute divergence to the related pair $A,B$ than to the unrelated random pair $C,D$.
>
> NTS is designed for a different role. It is bounded, rank-based, and correspondence-aware. Rather than comparing absolute topological divergence magnitudes, it asks whether topology-critical relations are consistently ordered under the sample correspondence. Therefore, although $A,B$ is a deliberately difficult related pair and its NTS value is not close to 1, NTS still correctly identifies that $A,B$ preserves much more paired structural organization than the randomly paired shell pair $C,D$.
>
> We also provide a controlled planted-hierarchy experiment in the supplementary material to address the reviewer’s concern about the decreasing trend of NTS in the Clusters experiment. There, an independent ground-truth structural distance is available, defined as $d_{\mathrm{GT}}=1-\frac{1}{L}\sum_{\ell=1}^L \mathrm{ARI}(P_\ell,\widetilde P_\ell)$, and we use $D_{\mathrm{NTS}}=(1-\mathrm{NTS})/2$. In this setting, $D_{\mathrm{NTS}\text{-}E}$ and $D_{\mathrm{NTS}\text{-}M}$ track the ground-truth trend strongly, with Spearman correlations 0.958 and 0.956 and pairwise ranking accuracies 0.912 and 0.910, respectively. The full code, visualizations, raw results, and detailed setup of both experiments are provided in the supplementary material.

---

> ### Author Response · Authors · 2026-05-01
>
> ## Q2: Sensitivity, nonlinearity, and relabeling behavior of NTS
>
> We thank the reviewer for raising this important point. We agree that the interpretation of Figure 4c should be clarified.
>
> NTS is not intended to be a linear effect-size measure with respect to the number of clusters. In the Clusters experiment, the number of clusters is only a control variable; it is not itself a linear ground-truth structural distance. By “normalized”, we mean that NTS is bounded and rank-based, not that its value should change proportionally to the cluster number or to any particular perturbation parameter.
>
> The sharp initial drop from one macroscopic group to two groups is expected in this construction. The point cloud is first formed from many local small clusters and then reorganized into 1, 2, 3, ..., 12 macroscopic groups. Therefore, the transition from 1 to 2 groups is the first global split of the point cloud, rather than a small local perturbation. This split changes many sample-wise neighborhood and merge-order relations at once. After this global reorganization, later changes from 2 to more groups are more incremental, so a slower change in NTS is expected. Thus, the curve should not be interpreted as saturation in the sense of losing all information; rather, it reflects that the first global split causes the largest reordering of topology-critical relations.
>
> We also added a controlled planted-hierarchy experiment to address this concern more directly. The original Clusters experiment does not have an independent ground-truth dissimilarity, so it is difficult to judge whether the exact decreasing rate is quantitatively “correct.” In the added experiment, we generate data from a known hierarchy and perturb a controlled fraction of samples across hierarchy leaves. The ground-truth structural distance is defined independently of all compared quantities as $d_{\mathrm{GT}}=1-\frac{1}{L}\sum_{\ell=1}^L \mathrm{ARI}(P_\ell,\widetilde P_\ell)$, and we compare it with $D_{\mathrm{NTS}}=(1-\mathrm{NTS})/2$. In this setting, $D_{\mathrm{NTS}\text{-}E}$ and $D_{\mathrm{NTS}\text{-}M}$ show strong agreement with the ground-truth trend, with Spearman correlations 0.958 and 0.956, respectively, higher than RTD and SRTD in this experiment. The detailed setup, code, and full results are provided in the supplementary material. This supports that the nonlinear behavior of NTS is not arbitrary: when an explicit structural ground truth is available, NTS-distance tracks the true structural trend well.
>
> Regarding the relabeling example, the key point is that NTS is a paired, correspondence-aware measure. It is invariant to a simultaneous permutation of the sample indices in both representations, but it is intentionally not invariant to permuting only one representation. In representation analysis, point $a$ in representation A and point $a$ in representation B correspond to the same input sample. Therefore, relabeling only representation B is not merely a harmless renaming; it changes the sample-wise correspondence used in the comparison.
>
> Under the reviewer’s one-sided relabeling, the unlabeled metric space of representation B may remain the same, and the number of connected components at a given threshold may also remain unchanged. However, the labeled neighborhood and merge-order relations have changed. For example, sample $a$ may be closer to $b$ than to $e$ in representation A, while after relabeling, sample $a$ in representation B may be closer to $e$ than to $b$. NTS is designed to detect exactly this kind of correspondence-aware change.
>
> Therefore, a change in NTS in this example is desired. It does not indicate a change in the unlabeled number of connected components. Instead, it reflects a change in the labeled, correspondence-aware topological organization of the same samples. In this sense, NTS is not a pure unlabeled topological invariant; it is a topology-aware representation similarity score for paired samples.

---

> ### Author Response · Authors · 2026-05-01
>
> ## Q3: Clarification of NTS-E versus NTS-M
>
> We thank the reviewer for pointing this out. In Equation 2, $T_w$ denotes the minimum spanning tree induced by the distance matrix $w$, and $\mathrm{path}_{T_w}(i,j)$ denotes the unique simple path connecting $i$ and $j$ in this tree. Therefore,
>
> $$
> m_w(i,j)=\max_{e\in \mathrm{path}_{T_w}(i,j)} w_e
> $$
>
> is the MST bottleneck value, i.e., the smallest threshold at which $i$ and $j$ become connected in the 0-dimensional filtration.
>
> NTS-E and NTS-M are computed on the same core edge set $C=E_w\cup E_{\tilde w}$, but they compare different quantities. NTS-E directly compares the edge distances $w_{ij}$ and $\tilde w_{ij}$ on $C$. NTS-M compares the corresponding merge times $m_w(i,j)$ and $m_{\tilde w}(i,j)$ induced by MST bottleneck paths.
>
> Thus, NTS-M can be viewed as a merge-time or hierarchy-level comparison, while NTS-E is an edge-level comparison. The map from direct distances $w_{ij}$ to bottleneck merge times $m_w(i,j)$ is a coarsening: different direct edge-distance orderings can induce the same merge-time ordering. Therefore, NTS-M can be less discriminative, whereas NTS-E can detect changes in direct topology-critical edge distances that do not necessarily alter the merge hierarchy.
>
> By saying that NTS-E is “stricter,” we do not mean that there is a pointwise numerical inequality between NTS-E and NTS-M for every pair of representations. Rather, we mean that NTS-E asks for agreement at a finer edge-distance level, while NTS-M asks for agreement after projecting these distances to MST bottleneck merge times. In particular, it is possible for NTS-M to be perfect while NTS-E is not, because the merge-time hierarchy can be preserved even when direct edge distances on $C$ are reordered.
>
> The two variants give the same signal when replacing each direct distance $w_{ij}$ by its bottleneck value $m_w(i,j)$ does not change the rank order on the core edge set $C$, and similarly for $\tilde w$. A simple special case is when every core pair under consideration is itself an MST edge, so that $m_w(i,j)=w_{ij}$ for those pairs.
>
> Overall, NTS-M provides the natural merge-time interpretation, while NTS-E provides a finer edge-level diagnostic.
>
> ## Q4: Interpretability of the “functional shift” in Figure 6
>
> We thank the reviewer for raising this question. The apparent discrepancy between CKA and the proposed topological measures is exactly the phenomenon we aim to highlight.
>
> The final pooling layer is computed from layer 8, but pooling is not designed to preserve the full fine-grained sample organization of layer 8. It is an averaging and aggregation operation that reduces spatial resolution and summarizes local activations. Prior work also supports this view: pooling layers reduce the spatial dimension of feature maps and are expected to retain useful information while discarding irrelevant local details (Gholamalinezhad & Khosravi, 2020). In the context of global average pooling, CAM-based analyses further show that pooling emphasizes discriminative class-relevant activation patterns for image-level prediction (Zhou et al., 2016).
>
> CKA can remain high because it compares sample-level Gram matrices. Average pooling is a spatial averaging operation on the layer-8 feature map: it removes spatial coordinates but keeps the channel-level average response. In the final layers of a classifier, class-discriminative information is largely encoded in channel activations, so the dominant class-level Gram structure can remain aligned after pooling.
>
> NTS and SRTD ask a different question. They depend on the ordering of pairwise distances, MST edges, and connected-component merging order. Average pooling can suppress spatial details and within-class variations, causing local neighbor relations and merge orders to be reorganized. Thus, high CKA and low NTS / high SRTD are not contradictory. They indicate that pooling preserves the dominant class-level Gram structure while changing fine-grained sample-wise topology.
>
> Therefore, the low NTS / high SRTD between layer 8 and the pooling layer does not mean that the pooling layer is unrelated to layer 8. Rather, it indicates a pooling-induced structural coarsening: the representation remains useful for coarse classification, but its fine-grained topology-aware organization is strongly reorganized. This is the specific functional shift we refer to.
>
> We also agree with the reviewer’s visualization suggestion and will synchronize the colorbar scales for NTS-E and NTS-M to $[-1,1]$.
>
> References:
>
> - Zhou, B., Khosla, A., Lapedriza, A., Oliva, A., & Torralba, A. (2016). *Learning Deep Features for Discriminative Localization*. CVPR.
> - Gholamalinezhad, H., & Khosravi, H. (2020). *Pooling Methods in Deep Neural Networks, a Review*. arXiv:2009.07485.

---

> ### Author Response · Authors · 2026-05-01
>
> ## Q5: Interpretation of RTD / Max-RTD asymmetry
>
> We thank the reviewer for the careful comment. We would like to clarify the intended interpretation of Table 4f.
>
> Table 4f is not intended to show a monotonic trend with the number of clusters. The key observation is a paired asymmetry pattern: when the directional asymmetry of RTD has a large magnitude, the directional asymmetry of Max-RTD also tends to have a large magnitude, but with the opposite sign.
>
> This pattern is consistent with the union/intersection interpretation of the three quantities. RTD measures a directional deviation relative to the union-like construction, while Max-RTD measures a complementary directional deviation relative to the intersection-like construction. SRTD directly measures the symmetric gap between the union-like and intersection-like constructions.
>
> More concretely, let
>
> $$
> A=\mathrm{RTD}(w,\tilde w),
> $$
>
> $$
> B=\mathrm{RTD}(\tilde w,w),
> $$
>
> $$
> C=\mathrm{MaxRTD}(w,\tilde w),
> $$
>
> $$
> D=\mathrm{MaxRTD}(\tilde w,w),
> $$
>
> and let $S=\mathrm{SRTD}(w,\tilde w)$. Empirically, the one-directional RTD plus the corresponding one-directional Max-RTD is close to SRTD, up to a small residual term. Writing this as
>
> $$
> A+C=S+2\epsilon_1,\quad B+D=S+2\epsilon_2,
> $$
>
> we obtain
>
> $$
> A-B=-(C-D)+2(\epsilon_1-\epsilon_2).
> $$
>
> Therefore, when the residual terms $\epsilon_1$ and $\epsilon_2$ are small, the RTD and Max-RTD asymmetries naturally have opposite signs and comparable magnitudes. This is the behavior observed in Table 4f.
>
> Thus, Table 4f should be understood as an empirical consistency check rather than a claim about cluster-number trends. The sign-opposed and magnitude-matched asymmetry supports the interpretation that the directional imbalance of RTD and Max-RTD comes from the same underlying union/intersection discrepancy, which SRTD captures directly.
>
> ## Q6: Use of SRTD in optimization
>
> We thank the reviewer for the helpful comment. We would like to clarify the intended role of SRTD in the optimization experiment.
>
> Our claim is not that an SRTD-based loss must numerically dominate RTD, Max-RTD, or RTD-lite on every validation quantity. The training objective is not validation SRTD alone: it combines reconstruction quality with a topology-preserving loss, is optimized on stochastic minibatches, and is evaluated on held-out validation data. Therefore, even when SRTD is used as the topology loss, it is not guaranteed to produce the smallest final validation SRTD among all methods.
>
> This is also expected because RTD, Max-RTD, SRTD, and their lite variants are closely related topological discrepancies. Optimizing one of them can reduce the others as well. Thus, RTD- or Max-RTD-based losses may obtain validation SRTD values comparable to the SRTD loss.
>
> To address the reviewer’s request, we provide checkpoint curves for validation RTD and validation SRTD from epoch $70$ to epoch $250$, evaluated every $10$ epochs, in the supplementary material as `val_rtd_srtd_panel.png`. The curves show that all topology-based losses steadily reduce both validation quantities, and that SRTD/SRTD-lite converge stably with performance comparable to RTD, Max-RTD, and RTD-lite.
>
> Thus, the optimization experiment should be interpreted as showing that SRTD and SRTD-lite are practical topology-preserving losses with comparable embedding quality, rather than as showing that SRTD must strictly dominate every RTD-style objective. Their practical advantage is the symmetric formulation and reduced computation, especially for SRTD-lite.

---

> ### Author Response · Authors · 2026-05-01
>
> ## Q7: What SRTD adds beyond RTD and Max-RTD
>
> We agree that SRTD can show scalar trends similar to RTD and Max-RTD in simple same-scenario perturbation experiments. This is expected, because these quantities belong to the same RTD-family of topological discrepancies. The contribution of SRTD is not that it must produce a qualitatively different curve in every setting.
>
> The main point is that SRTD provides a symmetric union/intersection comparison. RTD is based on a directional union-like construction, while SRTD directly measures the gap between the union-like and intersection-like constructions in a single symmetric object. This makes SRTD more natural as a symmetric divergence and also explains the complementary behavior of RTD and Max-RTD.
>
> This is not only a theoretical distinction. In the controlled ground-truth experiment, the symmetric versions are better aligned with the independent structural distance than their corresponding asymmetric/lite counterparts. SRTD is slightly better than RTD, while SRTD-lite improves substantially over RTD-lite:
>
> | Comparison | Spearman | Kendall | Pearson | Pairwise ranking accuracy |
> |---|---:|---:|---:|---:|
> | RTD $\rightarrow$ SRTD | $0.900 \rightarrow 0.908$ | $0.732 \rightarrow 0.743$ | $0.906 \rightarrow 0.914$ | $0.866 \rightarrow 0.872$ |
> | RTD-lite $\rightarrow$ SRTD-lite | $0.787 \rightarrow 0.891$ | $0.606 \rightarrow 0.721$ | $0.803 \rightarrow 0.899$ | $0.803 \rightarrow 0.861$ |
>
> This suggests that incorporating the intersection-like construction is empirically useful, especially in the lite setting. The improvement from RTD-lite to SRTD-lite is particularly large, indicating that the symmetric union/intersection comparison provides a more robust signal than the union-only lite comparison. This is also consistent with the Clusters experiment, where RTD-lite is less stable and can even give a less reliable trend, while the symmetric lite version behaves more robustly.
>
> Therefore, SRTD should be viewed as a symmetric, interpretable, and efficient member of the RTD family. Its role is to unify and improve RTD-style divergence analysis and to provide a practical symmetric optimization loss. It is not intended to replace NTS as the normalized cross-scenario similarity score; for heterogeneous cross-scenario comparison, we use NTS because RTD, Max-RTD, and SRTD remain unbounded absolute divergences.
>
> ## Q8: NTS–CKA discrepancy for DeepSeek and related models
>
> We thank the reviewer for pointing out this interesting discrepancy. We clarify the observations in Figures 7 and 8 separately.
>
> For Figure 7, NTS and CKA show broadly similar patterns for many models, but they differ for models such as DeepSeek, Llemma, LLaMA, and Baichuan. We do not use this figure to claim a model-specific mechanism. Rather, the point is that CKA and NTS emphasize different aspects of representation geometry. CKA measures global Gram/kernel alignment, whereas NTS measures rank-based topology-critical relations under sample correspondence. Therefore, a model can show weak global kernel alignment while still preserving part of its local or hierarchical organization, leading to a discrepancy between CKA and NTS.
>
> For Figure 8, we performed additional auxiliary checks for the DeepSeek cross-model discrepancy. We computed Orthogonal Procrustes similarity, RSA/RDM Spearman correlation, and kNN overlap, with the diagnostic figures provided in the supplementary material. These diagnostics support a more specific interpretation. DeepSeek has relatively low global linear alignability with other models under Procrustes, consistent with its low CKA values. This suggests that its global feature geometry is less linearly aligned with the others.
>
> However, the rank-based and local diagnostics show a different side. RSA/RDM Spearman correlation and kNN overlap indicate that DeepSeek still retains noticeable distance-rank and local-neighborhood similarity with several related models, especially the Qwen family. This is consistent with the NTS result: although DeepSeek is less aligned in global geometric or kernel structure, part of its rank-based and topology-aware organization remains preserved.
>
> Therefore, we do not claim that DeepSeek has identical topology to other models, nor do we make a causal claim about its training recipe. The conservative interpretation is that CKA/Procrustes capture global geometric or linear-alignment shifts, while NTS/RSA/kNN capture rank-based and neighborhood-level organization that can remain partially aligned. This supports the complementary role of NTS and CKA in representation comparison.

---

### Decision · Action_Editor_XbUJ · 2026-06-03

**Recommendation:** Accept as is

**Audience:**

Yes

**Audience Explanation:**

The paper is likely to be of interest to researchers working on representation learning, interpretability, neural network analysis, and evaluation of foundation models. The proposed metrics in the paper address practical limitations of existing topological representation-comparison methods, especially asymmetry, lack of normalization, and scalability and the ideas therein are potentially valuable to a meaningful segment of the TMLR audience.

**Claims And Evidence:**

Yes

**Claims Explanation:**

The reviewers broadly in agreement regarding their assessment. There is general consensus that the submission’s main claims are supported. The paper also provides theoretical justification for SRTD/SRTD-lite and NTS, and the empirical evaluation covers synthetic examples, CNN representations, and LLM representation comparisons. The authors' revision has addressed the main requests for clarification, including additional validation of NTS, interpretation of discrepancies with CKA, and discussion of SRTD's role relative to RTD variants. The remaining concerns are mostly about scope, such as reliance on 0-dimensional topology, which are not critical.